# Feasibility of a serious game coupled with a contact-based session led by lived experience workers for depression prevention in high-school students

Mandy Gijzen[1,2,3]*, Sanne Rasing[2,4], Rian van den Boogaart[2], Wendy Rongen[2], Twan van der Steen[4], Daan Creemers[2,5], Rutger Engels[3], Filip Smit[1,6]

1 Trimbos Institute (Netherlands Institute of Mental Health and Addiction), Utrecht, The Netherlands, 2 GGZ Oost Brabant, Boekel, The Netherlands, 3 Erasmus School of Social and Behavioural Sciences, Erasmus University Rotterdam, Rotterdam, The Netherlands, 4 Anxiety, Compulsion & Phobia Foundation; National Patient Organization, Driebergen-Rijsenburg, The Netherlands, 5 Behavioral Science Institute, Radboud University Nijmegen, Nijmegen, The Netherlands, 6 Department of Clinical, Neuro and Developmental Psychology and Department of Epidemiology and Biostatistics, Amsterdam University Medical Center, Location VUmc, Amsterdam, The Netherlands.

* MGijzen@trimbos.nl

**Data Availability Statement:** Data regarding the game Moving Stories have been added as

## Abstract

### Background

Stigma and limited mental health literacy impede adolescents getting the help they need for depressive symptoms. A serious game coupled with a classroom session led by lived experience workers (LEWs) might help to overcome these barriers. The school-based Strong Teens and Resilient Minds (STORM) preventive program employed this strategy and offered a serious game, Moving Stories. The current study was carried out to assess inhibiting and promoting factors for scaling up Moving Stories once its effectiveness has been ascertained.

### Methods

Moving Stories was offered in three steps: (1) introductory classroom session, (2) students playing the game for five days, (3) debriefing classroom session led by lived experience worker. Data was collected on the number of participating students, costs of offering Moving Stories, and was further based on the notes of the debriefing sessions to check if mental health first aid (MHFA) strategies were addressed.

### Results

Moving Stories was offered in seven high-schools. Coverage was moderate with 982 participating students out of 1880 (52%). Most participating students (83%) played the Moving Stories app three out of the five days. Qualitative data showed that the MHFAs were

Supporting information. Qualitative data: excerpts are within the paper.

**Funding:** Funding for the research project was provided by the municipalities of Asten, Deurne, Geldrop-Mierlo, Gemert-Bakel, Helmond, Laarbeek and Someren, The Netherlands. Moving Stories was funded by 'Het Stimuleringsfonds'. GGZ Oost Brabant and the Trimbos Institute provided the program materials. The funders had no role in study design, data collection and analysis, decision to publish, or preparation of the manuscript.

**Competing interests:** Moving Stories was developed by IJsfontein and Trimbos Institute in collaboration with Behavioural Science Institute of the Radboud University and 113 Suicide Prevention. Trimbos Institute has the exploitation rights of the Moving Stories. Trimbos Institute is a not-for-profit WHO Collaborative Centre for Mental Health. Trimbos Institute may licence third parties to use the Moving Stories intervention within routine preventive services. FS and MG are employees at Trimbos Institute, but have no share in licence revenues. This does not alter our adherence to PLOS ONE policies on sharing data and materials.

**Abbreviations:** LEW, lived experience worker; MHFA, Mental Health First Aid; RCT, randomized controlled trial; STORM, Strong Teens and Resilient Minds.

discussed in all debriefing sessions. Students showed great interest in lived experience workers' stories and shared their own experiences with depression.

## Conclusions

Bringing Moving Stories to scale in the high-school setting appears feasible, but will remain logistically somewhat challenging. Future implementation and scale-up of Moving Stories could benefit from improved selection and training of LEWs that played such an important role in grabbing the full attention of students and were able to launch frank discussions about depressive disorder and stigma in classrooms.

## Trial registration

The study is registered in the Dutch Trial Register: Trial NL6444 (NTR6622: https://www.trialregister.nl/trial/6444).

## Introduction

Adolescence is a critical period in developing affective disorders such as major depression [1]. Both depressive symptoms and depressive disorder are associated with a range of psychosocial difficulties first during adolescence such as lower educational attainment, social withdrawal, low self-esteem, possibly even suicidal thoughts and behaviors, and later in adulthood unemployment, financial debt, poorer marital and social functioning, and recurrence of depressive episodes [2]. Considering, the adverse impacts of depression, early recognition and preventive strategies are needed. Preventive strategies within the school setting have potential to reach most adolescents.

It is important that preventive programs for adolescents are embedded in a multimodal integrated stepped-care program for early detection and prevention of depression coupled with pathways to professional help. Indicated preventive strategies, that are aimed at adolescents with elevated depressive symptoms, were shown to have the largest effect sizes in depression [3–6]. However, universal preventive strategies, that target all adolescents regardless of risk or symptom level, have the added benefit that they reach most adolescents, may lower barriers related to receiving professional treatment and might be less stigmatizing [7]. A growing body of evidence suggests that integrating universal and indicated prevention in sequential stepped-care approaches with screening and monitoring of outcomes are most effective [8–10].

One of these multimodal prevention programs is the Strong Teens and Resilient Minds (STORM) program [11,12]. In the first iterations of STORM, screening was followed by an indicated CBT-based preventive program and was found to be effective in several RCTs [13–15]. Currently, the STORM program consists of four components: screening, gatekeepers' training of teachers, sequentially followed by universal and indicated preventive interventions that are coupled with pathways to professional mental healthcare. This iteration of the STORM program is currently being evaluated for effectiveness in an RCT [11]. It is worth noting that the gatekeepers' training, screening and indicated prevention program have been evaluated elsewhere and were found to be effective and feasible for future implementation [16–19]. The universal preventive strategy, *Moving Stories*, added to the latest iteration of the STORM program has not been evaluated for feasibility. Moving Stories in itself was evaluated

for effectiveness in a randomized trial and was found to reduce adolescents' personal stigma, and while symptom recognition (a component of mental health literacy) was also improved this effect was not significant compared to a control group [20]. Therefore, it is now important to determine whether Moving Stories can be successfully integrated and implemented as a new component within the STORM program.

The Moving Stories intervention was based on teen Mental Health First Aid (MHFA), an educational stigma intervention that was developed to improve mental health awareness [21]. The teen MHFA program was found to improve mental health awareness in adolescents, inspired confidence to support a peer with mental health difficulties, and increased personal help-seeking intentions, while reducing stigmatising attitudes [21]. This is important because depressive symptoms and suicidal behaviors are associated with decreased help-seeking due to fear of stigma and low mental health literacy [22–25].

A recent review concluded that adding direct social contact with a person with lived experience of mental health issues to educational stigma interventions was most effective in reducing mental-health-related stigma [26]. Another meta-analysis concluded that education alone and direct contact interventions were both effective in lowering stigma toward people with mental illness, but education was favored among adolescents [27]. Therefore, Moving Stories was based in both approaches: education as well as direct-contact with a lived experience worker.

Even if the reach of adolescents in schools can be large, the acceptability and uptake of universal preventive interventions could be further enhanced through use of gaming and smartphones. Over 90% of Dutch adolescents of age of 12 years and older own a smartphone [28]. Additionally, a study in the USA showed the importance of video games considering that 97% of their participants, aged 12–17, played video games [29]. A meta-analysis showed that using serious games for health promotion can lead to small, albeit positive effects in healthy life-style behavior [30]. Therefore, Moving Stories was designed as a serious game to be played by high-school students followed by a debriefing session led by lived experience workers.

In addition to the evaluation of effectiveness, it is also important to assess, and possibly improve, the feasibility of interventions for future scale-up. This requires the formulation of a well-tested implementation strategy that took into consideration the lessons learned from the intervention's implementation during the STORM trial. These lessons learned may guide future role-out of Moving Stories embedded in the STORM program. Acceptability of screening, gatekeepers' training and indicated CBT-programs have been reviewed previously as mentioned earlier [16–18]. Therefore, the primary aim of the current study was to assess the feasibility of implementing Moving Stories as a universal preventive intervention within the STORM program as this intervention was newly added within the STORM approach. More specifically, during the STORM trial we aimed to record (1) the number and percentage of high-school students that actively played Moving Stories (coverage rate), (2) the average number of playing days out of the five (participation rate), (3) how successful LEWs were in reviewing all five mental health first aid (MHFA) strategies during the debriefing sessions and to what extend students could identify depressive symptoms (fidelity), (4) what other themes were addressed and resonated with the students during the debriefing sessions, and how the students responded to the personal experiences of the LEWs. Finally, implementation costs were computed including the costs of recruiting and training LEWs for leading the debriefing sessions to see if money was spent wisely or requires reallocation. As recommended by Oakley et al. [31], assessing the feasibility of an intervention is performed before the effectiveness data of the STORM trial are analyzed.

## Materials and methods

### Participants and procedure

In the context of the STORM trial, the program of Moving Stories was implemented in two consecutive study years, namely 2017–2018 and 2018–2019. In the Netherlands the study year generally starts around September and concludes around July. Moving Stories was offered from January to May in each study year. Fifteen high schools participate within the STORM trial, of which seven were randomly assigned to the experimental condition. Moving Stories was only offered at schools in the experimental condition.

Nearly one thousand (N = 982) eight grade high-school students across seven schools in the Southeast of the province North-Brabant participated in Moving Stories. Data from the STORM trial indicated that 11–13% of these eight grade students were screen positive for depressive symptoms as measured by the Children's Depression Inventory 2, and 5–6% of them were screen positive on suicidal thoughts or behaviors.

Inclusion criteria were that they had to be enrolled in their second year of high school (i.e. 8$^{th}$ grade in the US) and were sufficiently fluent in the Dutch language to participate in class-room sessions (which were in Dutch). During their second year of high school most student in The Netherlands are 13–14 years of age. However, there are few exceptions and we did not want to exclude any students. Therefore, a wide age range of 11–15 years was used for inclusion.

### Intervention: Moving Stories

Moving Stories is intended to increase recognition of depression in self and peers, decrease stigma about depression and to promote help seeking behaviours. The intervention consists of a combination of a serious game and contact-based learning, comprising an introductory face-to-face lesson led by a researcher, single player game-app and a debriefing session led by a Lived Experience Worker (LEW) with experience of depression. The app Moving Stories was made available on the Apple App Store and Google Play app on 6 October, 2017 in both English and Dutch.

During the introductory lesson a researcher would explain to students how to download the app, what was expected of them and provided log-in codes. These sessions were delivered during a regular school class in a structured manner using a ready-to-use PowerPoint presentation to ensure every student received similar instructions. The aim of the game, as described to students, is that they will enter a virtual home where they will find a girl, Lisa, who is not feeling well. Students are asked to see if they can be of any help to Lisa. The term depression is not used during this introductory session. Although the game is played individually, the students are encouraged to help each other in identifying strategies that could be of any help to Lisa. Students were instructed to download the app during this introductory classroom session. After downloading the app, participants had to enter a passcode in order to receive access to the full app to ensure only adolescents from participating schools could actively use the app. As such, only participants that were enrolled in their second year of high school could participate.

Students then go on to play the game app for 10–15 minutes in the morning, between 7AM and 11AM, for 5 days. Students only had access to the app during the five days that they were asked to play, while they could still open the app students could no longer perform any actions or start over after the five playing days of their class was over. Students play the game individually, but simultaneous with their classmates and they are expected to do this in their own time. Schools did not provide the students with time to complete the game during school hours. In

the game students learn to interact with a virtual girl, Lisa, suffering from depressive symptoms. The students perform actions in the game (over 5 days) which consist of household tasks and/or offering Lisa help, and have the ability to talk to Lisa in a structured manner using pre-programmed options conversation). The aim is that they learn effective strategies when helping someone with depressive symptoms or to become more able to seek help for themselves. Students were prompted to play the game as reminders from the app would pop-up when Lisa's house would open (7AM) and an hour before the house would close (10AM). As with any mobile app, students had the ability to turn of pop-ups from the app. Feedback is provided each day by gaining or losing points indicating a *relationship score* indicating the quality of the relationship with Lisa based on the action they did in the game. In addition, they receive written text messages from Lisa telling the students how Lisa felt about each action they performed. In this way, students knew why they had gained or lost points. These messages were sent after school hours, from 3PM until 7PM.

After approximately seven days, students are brought back together again for the debriefing session. This debriefing session is an in-person classroom session lead by the LEW during a regular school class which are between 40 and 50 minutes long depending on the school. The LEW was aware of the length of each sessions beforehand. The LEW talks about his or her personal experience (approximately 10 minutes) and relates these personal experiences with depression to the game, which is then related to the students' experiences with the game. The LEWs are required to discuss the five first aid strategies from the teen Mental Health First Aid (MHFA): (1) look for warning signs, (2) ask how their friend or peer is doing, (3) listen without judgement, (4) help them connect with an adult, and (5) your friendship is important [21]. The five strategies are reviewed and wrapped up as take-home messages for the students during the debriefing sessions. For this the LEWs can choose to use the PowerPoint slides that were made available to them during their training. In the PowerPoint slides show some example questions to launch discussions about the five aid strategies, but the LEWs were free to use these slides and adapt it to their personal preference as long as the five strategies are discussed. Moreover, the structure of the debriefing session was deliberately left open so LEWs could adapt it to their personal preference and make impromptu responses to the themes brought up by the students.

In the current trial, a teacher trained as gatekeeper and a researcher were always present during both classroom sessions. However, for future implementation only a teacher would need to be present, but there is no need for an additional researcher to be present as well. A teacher could take over the role of the researcher in the first introductory classroom session. Teachers' presence is needed for the debriefing classroom session with the LEW as they are familiar with the students in the class and to ensure the LEW can focus on debriefing rather than keeping order in the classroom. Also, the teacher, trained as gatekeeper, will be able to guide students towards school mental health professionals. Further details about the content of Moving Stories and its development are described extensively elsewhere [32].

## Recruitment and training of lived experience workers

The LEW were recruited via a post on the Facebook page of the Trimbos Institute (Netherlands Institute of Mental Health and Addiction) and GGZ Oost Brabant (Mental Health Service). Additionally, we contacted several lived experience organizations. Requirements were that the LEW had some prior lived experience with depression. LEWs were informed during the recruitment process that they were required to tell their personal story to students in their second year of high school and that they would need to complete a training beforehand. Any LEW that applied was invited to participate in the training (n = 37).

The LEWs first had to complete an 8 hr training. This training for LEWs was designed by the authors, and a researcher who is also a LEW and was an experienced trainer. The training sessions were provided by these researchers as well. At the start of the training, participants were asked to play the serious game to fully understand what the students have done prior to their debriefing session.

The training itself consisted of a theoretical part and a practical part. During the theoretical part (3hrs) the LEWs learned more about the STORM program, its delivery and the goals of the Moving Stories intervention, and what their role was in the Moving Stories intervention. In the practical training (5hrs), the LEWs had the opportunity to practice telling their personal story in 10–15 minutes, and introducing the five MHFA strategies. The discussion of the five strategies from the MHFA was the most important component of the debriefing session and the LEWs were required to discuss all five strategies. Additionally, LEWs had to prepare themselves for questions that students might ask regarding their experiences, difficult topics (such as students having suicidal thoughts) and the game.

The LEW were provided with training materials (PowerPoints used during the training session) and a PowerPoint that they were free to use or adapt to their liking for the debriefing session. This PowerPoint consisted of a few example questions that the LEW could ask students as well as a slide with the five strategies from the MHFA.

The training sessions took place from September to December in both years indicating a time lapse of one to six months between training and the debriefing session. In the meantime the LEWs were updated on planning by the researchers via email and WhatsApp, but any other contact between the researchers and LEWs was minimal during the training and debriefing sessions. By design, LEWs always led the debriefing sessions in the presence of a gate keeper-trained teacher and a researcher, with the researcher taking notes.

## Data collection

Data was collected during the STORM trial. The medical ethics committee (Commissie Mensgebonden Onderzoek Region Arnhem-Nijmegen) in The Netherlands approved this study (NL61599.091.17). Informed consent from adolescents and their parents was obtained for the STORM trial. During the second trial year participation in Moving Stories was allowed on voluntary basis for students not participating in the STORM trial. At all times, students were informed that a researcher would be present during the debriefing sessions who might take anonymous notes.

Data that were collected are 1) data on preparing the delivery of Moving Stories, 2) quantitative data of offering Moving Stories, see Supplemental Materials, and 3) qualitative data on offering Moving Stories.

The data on preparing the delivery of Moving Stories included number of LEWs recruited and trained (and ensuing costs), number of trainings needed for classroom sessions, number of classroom sessions, and number of trained LEWs versus LEWs that actually provided debriefing session.

The quantitative data of offering Moving Stories included app usage data of the game and costs associated with debriefing sessions. App usage data was collected, anonymously, through a secure website. Data that were recorded include number of downloads, days played per student, and number of students playing each day. The app only collects fully anonymous data; players could not be identified as researchers were fully unaware of the identity of participating students and received only aggregated data.

In total, 49 debriefing sessions were provided by 11 LEWs. Notes were taken of 18 debriefing sessions to record what the LEW and students discussed during the debriefing sessions: 8

**Table 1. Overview of spread of schools and educational level used for minutes.**

| School | Nr of minutes taken | Educational level* |
|--------|---------------------|--------------------|
| 1 | 8 | 1 |
| 2 | 1 | 2 |
| 3 | 4 | 1 |
| 4 | 2 | 1, 2 |
| 5 | 1 | 1 |
| 6 | 1 | 1, 2 |
| 7 | 1 | 2 |

* 1 represents pre-vocational (in Dutch: VMBO), while 2 represents higher education/pre-university (in Dutch: HAVO and VWO).

during study year of 2017–2018 and 10 during 2018–2019 with at least one debriefing session per school (see Table 1). These notes were kept by four researchers to ensure that topics noted were not the results of reporter bias.

## Data analysis

Descriptive statistics such as means and standard deviations were computed in SPSS Version 27 [33]. With regards to the qualitative data from the notes, first we checked in how many debriefing sessions the required five aid strategies were discussed by the LEW. Additionally, we checked whether common themes arose across the different debriefing sessions. These themes were not pre-determined as there was no specific theme the LEW had to discuss with regard to their personal experiences or depressive symptoms in general. Instead, they were free to discuss several aspects of both the game and their personal experience with depressive symptoms and were encouraged to interactively tailor the session to the interests expressed by the students. Discussed topics were considered a theme if a topic occurred in at least two of the recorded notes. A researcher read and coded all notes using MAXQDA 2020 Version 2.4.0 [34]. Quotes representative for themes were then exported into Excel.

## Results

### Preparing moving stories

A total of 37 LEWs were trained divided over seven training sessions consisting of 6 participants on average. Overall, a total of 56 hours were spent on training sessions. The costs for each training are divided in costs of the trainer (hourly fee and travel expenses) and costs of the LEWs (travel expenses). The LEWs did not receive any compensation for time spent in training. However, opportunity costs in the Netherlands are valued at €15 per hr, indicating €4,440 total opportunity costs. There were no costs for renting the venue where training sessions were held. However, the costs of renting a training location are €350 per training, suggesting €1,400 for rent. The total costs of the trainer were €3748 and €50 for travel expenses. The travel expenses for the training of all the 37 LEW was €874 in total during the current trial.

### Offering moving stories

The introductory lessons took approximately 30 minutes per class. A teacher was present during these introductory sessions, so they were aware of what their students would be doing and

**Table 2. Percentage of total students that completed actions on consecutive game days.**

| Game days | Study year | | | |
|---|---|---|---|---|
| | 2017–2018 | | 2018–2019 | |
| | N | % | N | % |
| Day 1 | 204 | 60% | 454 | 71% |
| Day 2 | 200 | 59% | 421 | 65% |
| Day 3 | 190 | 56% | 425 | 66% |
| Day 4 | 173 | 51% | 362 | 56% |
| Day 5 | 162 | 48% | 339 | 53% |
| No actions completed during any day | 68 | 20% | 98 | 18% |

also to keep order in the classroom. For future implementation, a gate-keeper trained teacher would be expected to lead the introductory lesson. Teachers would need to be instructed first in order to provide these classes. This would take one hour and would cost €30–85. To ensure that each student was able to participate, regardless of smartphone possession, ten smartphones were purchased and made available for students at €1,750 total costs.

The total amount of students that were asked to participate in Moving Stories were 1,016 in study year 2017–2018 and 846 in study year 2018–2019 across the seven school locations. A total of 339 students downloaded the game during the study year 2017–2018 and 643 students during 2018–2019, indicating a coverage rate of 33.4% and 76% respectively across all second-year students. A number of students (n = 166, 16.9%) that downloaded the app did not complete any actions for the duration of their game in either year. It was noted that the number of students actively performing actions in the app gradually declined over the five consecutive days (see Table 2), but on average more than half of the participating students played the game on each of the five days. The average amount of days played by students was three days (M = 3.14, SD = .64).

In total, 49 separate debriefing sessions were provided by LEWs. Most sessions were provided by a single LEW. However, some LEWs preferred to provide classroom sessions with another LEW. This made them feel more comfortable as some of them were inexperienced in providing classroom sessions. They were matched with a LEW that did have prior experience. In total, four sessions were carried out by two LEWs instead of one. Not all trained LEWs were able to provide a debriefing session: 11 of 37 trained LEWs (29.7%) were able to provide debriefing sessions. Several LEWs indicated that classroom sessions interfered with other (volunteer) jobs. Most of them never responded to invitations to provide classroom sessions; and reasons for no-show remained largely unknown.

The costs for each classroom session were divided into an hourly fee and travel expenses for the LEW. Each LEW received an hourly fee of 30 euros for providing the debriefing sessions. The total costs for classroom sessions by LEW were €4,621, which included both the hourly fee and travel expenses. An overview of the costs in euros can be found in Table 3.

Discussing the five first aid strategies of the teen MHFA program were a mandatory part of the debriefing classroom session and LEWs were asked to discuss all five first aid strategies with the students. We checked in the minutes of the 18 sessions whether all five first aid strategies were indeed discussed. This was the case for all 18 sessions.

## Debriefing sessions

The notes that were taken during the debriefing sessions were also used to identify themes that were brought up by the students. It was noticeable from the classroom sessions that Lisa's (the

**Table 3. Overview of costs in euros.**

|  | Costs | Total Costs | % of total costs |
|---|---|---|---|
| Preparation |  | 4,672 | 42% |
| Trainer | 3,798 |  |  |
| LEW | 874 |  |  |
| Offering |  | 6,371 | 58% |
| Materials | 1,750 |  |  |
| LEW | 4,621 |  |  |
| Total costs |  | 11,043 |  |

main character in Moving Stories) behavior and responses triggered different reactions from students and captured many experiences from the LEW. Therefore, the themes that were discussed in classroom sessions were diverse, but it is worth noting that the LEW had the freedom to highlight how their experiences with depression was (dis)similar to Lisa's. Considering the heterogeneous nature of depression, we do not view this as a problem. Additionally, students' interests were different across sessions as well. For example, some classes were really interested in school pressure and perfectionism, whereas other classes talked more about suicidal behavior in depression.

Six themes that were identified during our review of the minutes are highlighted below. The order in which the themes are represented in this article does not reflect the order and in how many sessions these themes were actually discussed; the n represents the number of sessions in which these themes occurred. Below, the themes are ordered from the broader themes down to more specific ones. Moreover, there were several questions by students that were similar across debriefing sessions, which are also presented in theme 7. We have added quotes from students regarding several themes in italics: Quotes have been translated into English. The original Dutch quotes have been added in brackets for clarification.

**Theme 1: Recognition of depression and sad mood.**   During most classroom sessions (n = 15) it became clear that students recognized that Lisa was experiencing symptoms of depression. Students also were able to name several symptoms of depression, such as feeling sad, irritability, not wanting to do anything (apathy and feeling tired).

*"The game was about Lisa, who was depressed and you needed to help her." (in Dutch: "Het spel gaat over Lisa, die was depressief en moest je gaan helpen.")*

*"She was depressed. She did not want anything, she became angry if you made a phone call to school or opened the curtains." (in Dutch: "Ze was depressief. Ze wilde niks, ze werd boos als je school belde, of als je de gordijnen opendeed. . .")*

*"Then you feel worthless, think you cannot do anything and feel sad." (in Dutch: "Dan voel je je waardeloos, denk je dat je niets kunt en ben je verdrietig.")*

**Theme 2: Lisa's dismissive behaviors.**   Students often mentioned how Lisa was very dismissive in her reactions towards students when they tried to be helpful (n = 15). They mentioned how she did not like anything they did, even though they intended well. Some students said her reactions were unexpected and not very nice. Most LEW recognized this behavior in themselves and explained how, even though they knew people were trying to be helpful, couldn't bring themselves to respond in a less grumpy way. Often these themes linked up

nicely to discussing the first aid strategies. Specifically, it was a way for the LEW to discuss that asking, talking to them, and being a friend are more important than specific actions.

> *"I made her a toasted sandwich the first day, which she enjoyed, but the second day she didn't anymore. She didn't want anything anymore." (in Dutch: "Ik had de eerste dag een tosti voor haar gemaakt en dat vond ze lekker, maar de tweede dag ineens niet meer! Ze hoefde niks meer. . .")*

> *"It seemed like she didn't want any help, it was annoying that she never approved anything you did."(in Dutch: "Het leek niet of Lisa hulp wilde, het was irritant dat ze nooit iets goed vond wat je deed.")*

The following quotes from students illustrate how students realized that being in contact by talking to Lisa, being a friend and asking questions was more important than specific actions.

> *"If our relationship would have been better." (In Dutch: "als onze relatie beter was geweest."), in response to "What would have helped Lisa?" (in Dutch: "Wat had Lisa geholpen?".)*

> *"First, ask her (Lisa) if she was ready for you to call someone, that was the best way. Then, she didn't mind that much.". (In Dutch: "Eerst vragen of ze eraan toe was dat je iemand ging bellen, dat was de beste manier. Dan vond ze het niet zo erg.")*

**Theme 3: Students sharing personal experiences.** In half of the recorded debriefing sessions (n = 12) students shared personal experiences of knowing someone with a depression or suicidal behavior. This was another reason why it was very helpful to have a teacher present. The research employee and LEW that were present would not be available for any additional support after the classroom session ended, whereas the teacher would be. For this reason, teachers had received a gatekeepers' training as part of the STORM program in order to provide a safety net for students at their school.

**Theme 4: Help-seeking opportunities within school.** One of the five first aid strategies is that they should consider "helping your peer/friend connect with an adult". One important resource for students would be school, as this is where they spend most of their time. It was noticeable during several sessions (n = 10) that students were often unfamiliar with resources available at school or relied on outdated information (e.g., a counsellor who had left the school). In these cases it was very helpful to have a school teacher present during the sessions as they could provide the students with the correct information regarding counselors at school.

> *"I would first try to talk to a teacher, that feels less direct* [rather than going to my mom/ daddy]*." (In Dutch: "Ik zou er eerst met een leraar over praten, dat is minder direct.")*

> *"I know the school has a counsellor, but I don't know who he/she is. So, I would approach my mentor first." (In Dutch: "Ik weet dat er een vertrouwenspersoon is op school, maar weet niet wie. Ik denk dat ik dus eerst mijn mentor\* zou vragen.")*

\* Note: A mentor is a teacher who also acts as a trusted guide for students in matters related to their study. Typically, a mentor is assigned at the start of school year to either one or more classes, while a class typically has some 20–30 students.

Question asked by LEW: *"Does anybody know who you can approach to talk about depressive symptoms (red. for yourself or your friends)?"* (In Dutch: *"Weet iemand waar je naartoe kan hier op school als je met iemand wil praten over depressieve gevoelens?"*) This resulted in students mentioning several different names, but they seemed unclear who to turn to. The teacher then explained who the counsellors at school were.

**Theme 5: Physical health and mental health.**   In order to illustrate and stimulate help-seeking behaviors among students, a couple of LEW (n = 2) used a comparison to how one would seek help for a broken leg. Also, one LEW preferred the comparison with the flu for Lisa remaining at home instead of going to school. This seemed to resonate with students more, as they had their own experiences with flu or comparable physical illnesses which required them to stay home.

LEW: *"What if you had the flu and someone told you to get out of bed and pulled the blankets off you, how would you feel then?"* Student: *"I would not like that. If I have the flu, I just want to stay in bed and not talk to anyone."* LEW: *"When you are feeling depressed it is very similar. It would not help me to feel better if someone would do those things."* (In Dutch: *"Wat als je de griep had en iemand zie dat je maar uit bed moest komen en de dekens van je aftrok, hoe zou je dat vinden?", "Dat zou ik echt niet leuk vinden. Als ik de griep heb dan leg ik me in bed en praat ik met niemand.", "Als je depressie hebt dan is dat vergelijkbaar. Ik zou me echt niet beter voelen als iemand die dingen zou doen."*)

LEW: *"If you are feeling ill, you'd rather not get out of bed."* Student: *"So, is that the same as depression?"* LEW: *"Essentially yes, being forced out of bed or going outside does not help, unless you feel supported."* (In Dutch: *"Als je ziek bent, blijf je toch ook liever in bed.", "Is dat hetzelfde als wanneer je depressief bent?", "Op zich wel, als iemand je dwingt om uit bed te komen of naar buiten te gaan helpt dat niet, tenzij je steun van iemand krijgt."*)

**Theme 6: Animals/Pets.**   The students seemed to enjoy the presence of Lisa's cat in the house. It was mentioned or asked about in several sessions (n = 7), and often led to interesting discussions. It appeared that students really understood how the presence of a pet would help someone who is feeling sad. They mentioned how an animal could listen without judgement (it does not talk back at you).

*"Animals cannot offend you." (In Dutch: "Dieren kunnen je niet beledigen.")*

*"You can talk to them and they always listen." (in Dutch: "Je kunt er tegen praten en ze luisteren altijd naar je.")*

*"Dogs/cats cannot judge and do not get mad. They just come over to cuddle you." (In Dutch: "Honden/katten kunnen niet oordelen en worden niet boos. Ze komen gewoon naar je toe om te knuffelen.")*

**Theme 7: Frequently asked questions by students.**   Most question students asked were related to the personal experiences of the LEW (n = 10), illustrating that these experiences are interesting to adolescents and they do want to know more about it.

*"Are you feeling depressed right now?" (In Dutch: "Ben je nu depressief")*

*"How old were you when you first became depressed?" (In Dutch: "Hoe oud was u toen u depressief was?")*

*"What do you feel when you are feeling depressed?" "And how about when you are doing something you enjoy/like?" (In Dutch: "Wat voel je als je depressief bent?" "En wat als je iets leuks aan het doen bent?")*

*"How long have you been feeling depressed?" (In Dutch: "Hoe lang bent u al depressief?")*

*"Does it help to provide classroom sessions?" (In Dutch: "Helpt het als je voor de klas staat?")*

Many questions were also related to the parents and the LEW relationships with their parents (n = 6).

The following quotes illustrate the questions students asked regarding the LEW's relationships with their parents:

*"How is your relationship with your parents now?" (In Dutch: "Hoe is de relatie met je ouders nu?")*

*"How are your parents dealing with you being depressed?" (In Dutch: "Hoe gaan je ouders ermee om?")*

*"Does your mother know that you are depressed?" (In Dutch: "Weet je moeder dat je depressief bent?")*

## Discussion

The aim of the current study was to assess the coverage, costs of preparing and offering the Moving Stories intervention across seven high-schools in the Netherlands, the fidelity with which the living experience workers (LEWs) reviewed all five mental health first aid (MHFA) strategies and to identify any additional themes that were brought up by the participating high-school students during the debriefing sessions. This was done to find areas where the implementation of the Moving Stories intervention could be improved and the feasibility of future scale-up of Moving Stories enhanced.

### Principal findings

Among the 1880 eligible students 982 (52%) downloaded the Moving Stories app and 83% of them played the game for at least one day with an average of three days out of the five days. The LEWs discussed the five MHFA strategies with students in all recorded sessions. The notes of the debriefing sessions indicated that sadness-like symptoms were readily recognized as indicative of depression, whereas dismissive or grumpy behaviors were not. The students showed great interest in the personal stories of the LEWs and in more than half of the debriefing sessions one or more students shared their own personal experiences with mental health and depressive symptoms. This may suggest that mental health literacy was enhanced and stigma reduced among students. The total cost of implementing Moving Stories across seven high schools was €11,043 (US$ 12,367), of which €4,672 (US$ 5,232) was spent on the training

of the LEWs. However only 11 out of the 37 trained LEWs (30%) carried out their training in practice and led a total of 49 debriefing sessions. This suggests that selection and training of LEWs should be improved.

**Preparing Moving Stories.**   Recruitment and training of 37 LEWs costed €4,672 ($5,232) or €126 ($141) per LEW. The total costs (including recruitment and training costs of LEWs) of offering Moving Stories to 7 high schools was €11,034 ($12,357) or €1,576 ($1,765) per school. Since 982 students participated in Moving Stories, this amounted to €11 ($13) per participating student. This was achieved below the planned budget ceiling. However, it should be noted that of the 37 trained LEWs, only 11 led debriefing sessions, which suggests that recruitment and training of the LEWs should be improved.

For this project, every LEW who indicated they wanted to be involved in the project, was allowed to complete the training. Thus, it would be beneficial to develop a more specific set of criteria for selecting a LEW and make it very clear what is expected of LEW. Selection criteria that we would recommend are that the LEW has some prior experience in delivering their personal story to a public, preferably with high school students in classrooms. Recruitment of LEWs via lived experience organizations could help finding qualified LEWs.

The high drop-out rate among the LEWs further suggests the need to more intensively involve the LEWs in the design of the debriefing session and how it is implemented. After all, their engagement and sense of ownership may hold the key to their willingness to turn the debriefing sessions into a success. Ultimately, a sense of belongingness is important [35]. In addition, the training session for the LEWs may need to be better geared towards acquiring the ability and confidence to provide debriefing sessions in the classroom setting [36]. We refer to research by Chen et al. [37] who identified four main activities that are needed to adequately prepare a LEW for contact-based interventions. First, LEWs need to be psychologically prepared: they need adequate support and confidence. Second, adequate knowledge and skills need to be built. Third, a LEW needs to develop their personal story in a way they can share it. Fourth, and most importantly, the LEWs need to be able to practice the entire session.

Therefore, we recommend more time is allocated to the practical part of the training such that each LEW has ample training time to develop and tell their personal story in 10–15 minutes, and where the LEW can practice how to introduce the five MHFA strategies and learns how to anticipate to the various reactions and questions that students might bring up in real debriefing sessions. As a case in point, a prior study found that a transient dip in one's mood could occur within hours or first couple of days after having shared one's personal experience related to mental—although in the long run one's mental health will not be affected and LEWs may feel that sharing their story was beneficial to themselves and others [35].

The high drop-out rate among the LEWs may also indicate that the level of contact between the research staff and the LEWs in the period between training and the debriefing sessions was too low. This suggests another area where improvement can be made. Unfortunately, the current study did not ask the LEWs how the level of contact with the researchers affected their engagement in the implementation of Moving Stories. For future evaluations, it is important that the experiences of LEW are evaluated to determine which aspects or processes might impede further participation especially during the implementation.

**Offering Moving Stories.**   Overall, the coverage rate was moderate as a little over half of eligible adolescents downloaded the game (53%). Once an adolescent had downloaded the game the uptake of the program for adolescents seemed quite high, considering most students playing more than half of the game days and few students who downloaded the game did not play. This is remarkable bearing in mind that students had to play the game-app outside of school hours in their own time and in addition to their regular homework. Notably, during the second year more students played the game overall, and more students also played each

individual day than during the first year of implementation. The larger uptake rate could be explained by two things. First, during the second year a broader inclusion was allowed; students not participating in the STORM trial were still allowed to play Moving Stories when they so wanted. Students were encouraged to help each other in identifying strategies that could aid Lisa. As such, if more students are playing, more students are asking each other for help and in turn encouraging and enthusing each other to play. This self-reinforcing process could explain the difference in player frequency of each individual game day between the first and second year. Second, adolescents model their behavior mostly after peers [38,39] as to avoid the risk of social rejection [40]. Accordingly, they are more likely to participate if more peers are engaged.

A potential facilitator for increasing uptake and coverage rates would be to fully embed Moving Stories in the school curriculum and make it possible for students to play the game during school hours (playtime is approximately 10–15 minutes per day). This would ensure that students have time to play the game, but would also underscore the importance of the game when it is part of the curriculum. Such an approach would also increase normalization of mental health education at school, but would require collaborative planning and coordination of the STORM program with schools.

**Debriefing sessions.** During the debriefing sessions it became clear that the students recognized Lisa's depression symptoms, which is in accordance with previous studies [41–43]. On the other hand, a study by Burns and Rapee [44] found that adolescents were less likely to correctly identify a vignette without the obvious symptoms of sadness as having depression. This is consistent with our findings that Lisa's dismissive and grumpy behavior was less readily recognized as part of her depression, because irritable behavior is not an obvious and well-known symptom of adolescent depression. As such, it is important to include these types of less obvious symptoms as well, rather than just focus on "sadness-like" symptoms. This is how depression is generally portrayed in media, but depression is much more heterogeneous than that. Considering the link between mental health literacy and help-seeking behaviors [41,45], it is important that adolescents have knowledge of symptoms such as irritability as part of depression in order to ask or look for help. This is especially the case for adolescents as these symptoms are considered key symptoms of adolescent depression and not in adult depression [46].

It was clear from the notes from the debriefing sessions that students were willing to share personal experiences of friends and family with depression as well as their own experience with depression or other mental health difficulties. This indicates that there was a sense of trust and safety during these sessions where students felt comfortable enough to share their own stories and experiences. Moreover, this helped a number of young people realize that they were not alone with these experiences. This, in turn, could help to reduce the fear of stigma attached to depression as one could see that peers with depressive complaints were still participating in school and normal activities. Previous research has indicated that an reciprocal exchange in an open and safe space is vital in sharing personal experiences [35]. As such, by first letting the LEW talk about their own experience, they create such an environment of reciprocity, making it easier for students to share their own stories. Moreover, when students feel comfortable to share their personal experiences with mental health, they are more likely to seek help [47].

This underscores the importance of a teacher, who is trained as a gatekeeper, being present during the debriefing sessions. In the STORM program this was assured by providing gatekeepers' training for suicide prevention to teachers. Additionally, it could be beneficial that the LEW is accompanied by a mental healthcare professional (i.e. a clinician). Furthermore, it is advisable that schools consider embedding universal prevention within a larger stepped

prevention program considering students sharing such personal stories during these sessions. To illustrate, within the STORM program this was assured by offering students with an elevated level depressive symptoms a free indicated CBT-based preventive program, while also other pathways towards professional care were made available to the schools and their students.

The themes that really resonated with the students were the topics they could relate to or recognize in their daily life. Most students could recall a day they stayed home from school due to a cold or flu, and could see how a pet would be helpful owing to their own experiences with pets. Thus, it is important to consider relating complex issues such as mental illness to common experiences.

In similar vein, the students were especially interested in the LEW's current and past relationship with parents. Clearly, in this age group, parents play a large part in their social interactions, but at the same time adolescents are trying to become more independent from their parents [48,49]. Yet, the most common mentioned help-seeking strategy among adolescents is to turn to their parents [50]. Thus, involvement of parents in school-based depression programs could be beneficial [51].

## Strengths and limitations

A strength of the study is that at each participating school at least one debriefing session was monitored to reduce selective sampling bias and to enhance external validity. Moreover, notes of the debriefing sessions were kept by four different note takers to reduce bias.

Several limitations are important to note. The study was carried out in a small region in the Netherlands (the South-East region of Brabant), and may therefore not be generalizable to other regions and countries.

The debriefing sessions were not tape-recorded and transcribed verbatim. Tape-recording would have legal implications as participants would be identifiable. Moreover, tape-recording has several disadvantages, namely when discussing personal issues, participants are less likely to discuss these freely [52,53]. Additionally, the recording of the notes was not completed in a pre-determined standardized format. This can be problematic as it could be that the minutes are somewhat skewed to outcomes deemed important by the reporter [54]. However, the usage of multiple reporters may counter-balance reporter bias. Moreover, using an unstructured form also meant that possible unexpected topics were recorded.

Furthermore, the current study did not tap into the experiences of the students, the LEW, teachers and school administrators on how they experienced the Moving Stories program. Questions of interest could include how they rated the game and classroom sessions, and whether they would play the game again, or would recommend it to a friend or colleague. This would help to determine the acceptability of the intervention. With regards to the acceptability another study found that over 90% of students would recommend the Moving Stories game and 66% would recommend the debriefing session to a friend [20].

## Conclusion

Concluding, implementation of Moving Stories during the STORM trial to prevent depressive symptoms in high-school students was feasible. However, there were few trained LEWs that were able to provide debriefing sessions suggesting that selection and training of LEWs needs to be improved. The role of the LEWs is important because they managed to grab the full attention of students and were able to launch frank discussions about depression and create a space wherein students were willing to share their personal experiences. This suggests that

Moving Stories was successful in enhancing mental health literacy and reducing depression stigma.

## Supporting information

**S1 Checklist. Standards for reporting implementation studies: The StaRI checklist for completion.**
(DOCX)

**S1 Dataset. Dataset moving stories playing days.**
(SAV)

## Acknowledgments

We would like to acknowledge Rian van den Boogaart (project manager at GGZ Oost Brabant) for her great contribution to design, practicability and execution of the study. Additionally, we would like to thank Wendy Rongen, Bianca van Hulst and Esma Ballafkir (the research assistants at the time) for their help in the data collection. Also, we want to acknowledge Anouk Tuijnman (PhD student at Radboud University/researcher at Trimbos Institute) and IJsfontein for developing Moving Stories. We are also grateful to the collaborating schools (Alfrinkcollege, Carrolus Borromeus College, Commanderijcollege, Dr. Knippenbergcollege, Jan van Brabant College, Hub van Doorne, Peellandcollege, St. Willibrord Gymnasium, Strabrecht College, Vakcollege Helmond, and Varendonck College), the health professionals of the Municipal Health Services "Brabant-Zuidoost", mental health professionals of GGZ Oost Brabant, Marianne van Bakel for training the LEW, and the LEW providing the debriefing session for making this research possible.

## Author Contributions

**Conceptualization:** Mandy Gijzen, Sanne Rasing, Filip Smit.

**Data curation:** Mandy Gijzen, Wendy Rongen.

**Formal analysis:** Mandy Gijzen.

**Funding acquisition:** Sanne Rasing, Daan Creemers, Rutger Engels, Filip Smit.

**Investigation:** Mandy Gijzen, Rian van den Boogaart, Wendy Rongen.

**Methodology:** Mandy Gijzen, Sanne Rasing, Filip Smit.

**Project administration:** Mandy Gijzen, Sanne Rasing, Rian van den Boogaart, Wendy Rongen.

**Supervision:** Sanne Rasing, Daan Creemers, Rutger Engels, Filip Smit.

**Writing – original draft:** Mandy Gijzen, Sanne Rasing, Filip Smit.

**Writing – review & editing:** Mandy Gijzen, Sanne Rasing, Rian van den Boogaart, Wendy Rongen, Twan van der Steen, Daan Creemers, Rutger Engels, Filip Smit.

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
