## [Decision Letter · Decision Letter 0]

6 Jul 2021

PONE-D-21-17344

Implementing a serious game led by lived experience workers for depression prevention in high-school students: A process evaluation.

PLOS ONE

Dear Dr. Gijzen,

Thank you for submitting your manuscript to PLOS ONE. After careful consideration, we feel that it has merit but does not fully meet PLOS ONE’s publication criteria as it currently stands. Therefore, we invite you to submit a revised version of the manuscript that addresses the points raised during the review process.

We look forward to receiving your revised manuscript.

Kind regards,

Alison L. Calear

Academic Editor

PLOS ONE

Journal Requirements:

"Funding for the research project was provided by the municipalities of Asten, Deurne,

Geldrop-Mierlo, Gemert-Bakel, Helmond, Laarbeek and Someren, The Netherlands.

Moving Stories was funded by ‘Het Stimuleringsfonds,’ and OVK was funded by

ZonMw. GGZ Oost Brabant and the Trimbos Institute provided the program materials.

The funders had no role in study design, data collection and analysis, decision to

publish, or preparation of the manuscript."

We note that one or more of the authors is affiliated with the funding organization, indicating the funder may have had some role in the design, data collection, analysis or preparation of your manuscript for publication; in other words, the funder played an indirect role through the participation of the co-authors. If the funding organization did not play a role in the study design, data collection and analysis, decision to publish, or preparation of the manuscript and only provided financial support in the form of authors' salaries and/or research materials, please do the following:

a) Review your statements relating to the author contributions, and ensure you have specifically and accurately indicated the role(s) that these authors had in your study. These amendments should be made in the online form.

b) Confirm in your cover letter that you agree with the following statement, and we will change the online submission form on your behalf:

"Moving Stories was developed by IJsfontein and the Trimbos Institute in collaboration

with the Behavioural Science Institute of the Radboud University and 113

Zelfmoordpreventie.

Trimbos Institute has the exploitation rights of the Moving Stories. Trimbos Institute is a

not-for-profit WHO Collaborative Centre with the goals to disseminate best and

evidence-based practices. Trimbos Institute may licence third parties to use the Moving

Stories intervention within routine preventive services. FS, and MG are employees at

Trimbos Institute, but will not have a share in any possible licence revenues.

OVK2.0 was adapted by GGZ Oost Brabant in collaboration with the Behavioural

Science Institute of the Radboud University and Trimbos Institute and is published by

GGZ Oost Brabant."

Reviewers' comments:

Reviewer's Responses to Questions

**Comments to the Author**

1. Is the manuscript technically sound, and do the data support the conclusions?

Reviewer #1: No

Reviewer #2: Partly

2. Has the statistical analysis been performed appropriately and rigorously? 

Reviewer #1: N/A

Reviewer #2: N/A

3. Have the authors made all data underlying the findings in their manuscript fully available?

Reviewer #1: No

Reviewer #2: No

4. Is the manuscript presented in an intelligible fashion and written in standard English?

Reviewer #1: Yes

Reviewer #2: Yes

5. Review Comments to the Author

Reviewer #1: This study presents a process evaluation of a large-scale study of a serious game delivered in secondary schools by a lived experience worker. The inclusion of people with lived experience in the delivery of the program is positive; however, a critical problem with this study, is that it states that it intends to do a process evaluation, but it appears to not be measuring the rollout of the program against a specific process to assess how effective it was. The study presents more as a feasibility study, but this is also problematic as not all groups were included as participants so that the true feasibility of the program can be appropriately evaluated e.g., the discussion makes some important points about how to recruit and support lived experience workers, but this information does not seem to have been gained through the current study results This is because the lived experience workers were not asked about their involvement in the program, e.g., what was difficult for them, and what could have helped – instead the authors rely on assumptions. This seems like a large oversight for the conclusions the authors wished to reach, and is also noted as a limitation – “Furthermore, the current study did not directly tap into the experiences of the students, the LEW, teachers and school administrators on how they experienced the Moving Stories program”. I think perhaps a reframing of the study would help, as I don’t believe that currently this study supports the “feasibility” of the program as stated in the conclusions because it did not tap into those critical experiences.

My specific comments to the authors are below:

Abstract

1. The abstract should be more specific about what was done and what was found – do not list what you will present in the paper later on, this information should be presented here as well. At the moment, the methods includes a description of the Moving Stories program, but not what was actually done – were there any participants, list these here, how was the data for the results collected? Also, more information should be given in the results – it says that “28.9% of trainees were able to carry out their training” – how many trainees were there overall?

Introduction:

2. “Depression symptoms, if ignored, are likely to develop into major depression disorder [1]” I believe this article is trying to demonstrate that the experience of major depression in childhood, increases the risk of depression in adulthood; thus, I would suggest revising the sentence.

Methods:

3. Participants: The age range seems very broad (11-15 years) for only 8th grade students – can you please clarify why this was the case?

4. Important to confirm if the ethics statement listed at the beginning in the table for the journal will be included in the final article, as otherwise I would list this here.

5. Procedure: Could you explain in more detail how the program was delivered? How many schools were involved, how many teachers, how many lived experience workers? This sort of information seems important for a process evaluation, it should be included in the methods (not just the results) and should also be included in the abstract where indicated above.

6. Did students have access to the program from January to May? Or just over the five days? If this could be described a bit more clearly that would be helpful.

7. What specific data analyses were conducted? List these here under a heading: Data analysis.

Results

8. The description of the training required (8 hrs for LEWs) should be included in the method, and then the cost analysis of this can be the results.

9. Again, the information listed here about the number of schools etc., needs to be in the methods. The results can remain, but it is important to list who was involved (participants) and what you did (procedure) in the methods first.

10. How were the themes for the ‘debriefing sessions” identified - if this qualitative analyses, this needs to be explained and listed in the method? Also what was the purpose of analysing this information – was it to use it to improve the program? Or just out of interest?

Discussion

11. Could you give an overview of what was found under the principal findings heading, it would be helpful to really identify what the key findings were and list them here.

12. There are some critical statements that are made in the discussion, which seem to overstep the findings of the current study. For example, it is not clear where the data that inform the statement “it is important that each person who has a role in the program feels involved in the program and is motivated to bring the program to success” – or “it is important that each person is aware of the program components” how do you know this, what data was uncovered in your study that tells you this is the case? There is very little in regard to information collected from either the schools or the LEWs, so it is difficult to see where these statements have come from.

13. “A more efficient method of selecting the LEW needs to be developed before LEW are offered a training as part of their involvement in the Moving Stories intervention” – what does this mean and how did you come to this conclusion from your data? Did you ask LEWs about their involvement and any barriers they faced? There is a line in the results that says that some indicated that it interfered with their work, but it is not clear if this information was collected as data for the current project (e.g., were LEWs participants in the study too?).

Conclusion

14. I am not sure that you can say the feasibility of the program has been demonstrated via this study. If anything, there seemed to be issues with the LEWs and their participation and there is no data collected from the schools about how it fit into the curriculum etc., so it is unclear how this statement was reached based on the data presented.

Reviewer #2: Manuscript title: Implementing a serious game led by lived experience workers for depression prevention in high-school students: process evaluation

General comments: This manuscript describes a process evaluation of a high-school-based universal depression prevention program, which combines a serious smartphone game and an in-person discussion session with a lived experience worker. Some interesting reflections are provided on the challenges of training and maintaining lived experience workers, and the successes of generating classroom discussions through a debriefing session delivered by a lived experience worker. However, greater clarity around the objectives, methods and interpretation of findings is needed. The manuscript may also benefit from incorporating some discussion of how the lessons learned from implementing Moving Stories can be applied to other similar programs. The manuscript is generally clearly written. While I have noted some points of ambiguity in my specific comments, I do recommend the authors review the manuscript for any typographical or grammatical errors that are not specifically noted during review. I have made a number of specific comments below, I hope they are helpful to the authors in revising the manuscript.

Specific comments:

Title and Abstract: Moving Stories is characterised as a serious game led by lived experience workers. This seems misleading, as the only lived experience involvement described in the manuscript is the role of the lived experience workers in delivering a debriefing session. This element of the program was delivered by people with lived experience, but there is no evidence of true lived experience leadership. Please clarify whether this program involved lived experience leadership (i.e., the game was developed by people with lived experience, the program was developed and run by people with lived experience). If this is not the case, please adjust the title and abstract to accurately represent the nature of the Moving Stories program.

Introduction:

Page 6, Lines 109-110: What is meant by “a game-based design was used for Moving Stories led by lived experience workers.” Was the design and implementation of the program led by lived experience workers, or does this refer to the delivery of the briefing session? Please ensure the level of lived experience involvement is accurately characterised.

Page 6, Lines 117-118: What were the specific objectives of the process evaluation? What elements of program implementation did you plan to evaluate? Outlining these objectives in the Introduction section would provide more context for the methods and findings presented below. At the moment it is somewhat unclear why certain kinds of data were collected (e.g., costs) and others were not (e.g., perspectives of lived experience workers, school staff, and students on acceptability/feasibility).

Materials and Methods

Participants and Procedure, Page 6: How many schools were involved in the trial? Were different types of schools involved (e.g., government vs private funded)? What level of diversity did schools represent (e.g., average socioeconomic status of student population, range of geographical areas)?

Participants and Procedure, Page 7: How were students recruited? How did they provide informed consent (for app and research)? Were students required to participate in the research in order to access the Moving Stories program and participate in the debriefing session?

Intervention description, Page 7, Line 148: “Lived Experience Worker of depression” doesn’t quite make sense. I’d suggest changing this to “Lived Experience Worker with experience of depression” (or something similar) throughout the manuscript.

Intervention description, Page 7, Lines 149-150: This section should specify that the introductory lesson was led by a researcher. Additionally, please specify what information students were given in the introductory session (i.e., what were they told the aim of the program was? Is depression specifically mentioned?).

Intervention description, Pages 7-8, Lines 151-155: Some elements of program delivery need clarification. When did students complete the game – in class or in their own time? Were they prompted to play the game each day, if so how? Why could the game only be played for a limited time period in the morning, particularly if the game was played outside of class?

Intervention description, Page 7, Lines 160-163: How long was the debriefing session? Aside from the requirement to cover the five MHFA strategies, were there any other guidelines or requirements for the structure of the session?

Data collection, Page 9, Line 183: Information about the cost of the Moving Stories app was not reported. Please add this to the Results or remove this statement from the Methods.

Data collection Page 9, Line 188: Costs associated with debriefing sessions are already listed in the paragraph above, statement can be removed from this paragraph.

Data collection, Page 9, Lines 191-193: Was informed consent obtained from students and lived experience workers permitting the collection and reporting of session minutes? This is particularly important as individual students and workers have been quoted in the Results section.

Data collection, Page 9, Lines 191-193: The spread of recorded sessions across schools should be noted here.

Data collection, Page 9, Lines 194-199: Please describe the analysis methods used to produce themes from the debriefing session minutes.

Results:

Preparing delivery, Page 10: How were Lived Experience Workers recruited and selected? Were they required to have any previous experience? How was the training program developed? Was the training program developed with any lived experience input, or input from organisations with experience in delivering lived experience content in classroom settings? Who ran the training sessions? Were any materials provided to training participants for future reference/revision? Given the focus of the Discussion and Conclusion on the need to improve training and support of Lived Experience Workers, it is important to provide these details.

Preparing delivery, Page 10: The utility of reporting costs is unclear. Was a cost-benefit analysis conducted? How do these costs compare to other programs? From the information presented, it is not possible for the reader to determine whether the program represented good value for money.

Offering Moving Stories, Pages 10-11: Again, the utility of reporting costs is unclear. If costs are to be reported, perhaps summarising costs across preparation and implementation in a single table would clarify the total cost of running the program?

Offering Moving Stories, Page12, Lines 250-259: Is any information available about how Lived Experience Workers were communicated with and kept engaged with the program between training and classroom sessions? This may help shed some light on possible areas for improvement when next implementing the program.

Debriefing sessions, Page 13: Beyond the MHFA topics, was there any guidance on the structure of the debriefing sessions? This may have influenced what was most commonly discussed. It is important to outline the structure of the debriefing sessions in the Methods.

Debriefing sessions, Page 16, Line 359: Here and in a few other places in the manuscript, school mentors are referred to. What is a school mentor? This term needs defining for an international audience.

Discussion:

Program costs are not discussed in the Discussion section. This makes me unsure of the utility of reporting them – what was the purpose of assessing costs? What do the costs mean in practical terms?

Preparing Moving Stories, Pages 20-21, Lines 450-461: While some interesting points are made here, it is unclear how much of this content relates to the findings reported in the Methods section. Please clarify the links between these assertions and your process evaluation data. Additionally, this section feels like it is missing citations – please add in citations to reference materials as appropriate.

Preparing Moving Stories, Pages 21-22: Parts of the discussion around lived experience workers feels like it is focused on perceived weaknesses of the workers themselves (e.g., they may not be psychologically ready to work in a classroom), rather than focusing on elements of the program and its implementation that could be improved to better support a lived experience workforce. I do not think this is intentional, however, I think it would be beneficial to review the discussion points around the lived experience workers to ensure they are focused on evaluation findings about the program, rather than assumptions about the workers. It may also be helpful to highlight the lack of data on Lived Experience Worker experiences in these discussions – it is hard to propose effective support strategies without knowing what workers enjoyed about delivering the program and what they would change.

Preparing Moving Stories, Page 21, Line 465: Is efficient the best word to describe the improved recruitment strategy? Is ‘specific’ or ‘selective’ a more accurate term for what is meant here?

Preparing Moving Stories, Page 21, Lines 482-484: Was this a limitation of the implementation of recruitment and training – was the Lived Experience Worker role and what it involved inadequately explained before people volunteered for training?

Preparing Moving Stories, Page 22, Lines 487-496: This seems like a relevant place to discuss costs, if they are to be better incorporated into the manuscript.

Offering Moving Stories, Page 22, Lines 502-503: Please note this in the Methods section.

Offering Moving Stories, Page 22, Lines 506-509: Please note this context around difference in app access and encouragement of students to help each other play in the Methods section.

Offering Moving Stories, Page 23-24, Lines 529-538: Parts of this paragraph are covered in the Preparing Moving Stories section, perhaps it could be moved up and integrated with other Lived Experience Worker-related topics? Additionally, were there any indications from the evaluation data that a health professional would make Lived Experience Workers feel more comfortable? Or that workers were worried about the impact of delivering the session on their mental health? Making assumptions about the mental state of people who chose not to participate in delivering sessions is potentially stigmatising. As noted above, I recommend refocusing this discussion point to emphasise findings from the evaluation (e.g., students tend to share personal stories during debriefing and may thus benefit from direct access to a health professional, lived experience workers may feel more comfortable working in pairs), rather than assumptions about the Lived Experience Workers.

Debriefing sessions, Page 24, Lines 552-553: Citation for this sentence?

Debriefing sessions, Page 25, Lines 564-565: Citation for this sentence?

Strengths and Limitations, Page 26, Lines 586-588: This information should be noted in the Methods section.

Recommendations, Page 26, Line 608: While it is an interesting point, I am unsure how the data support the recommendation that partners in the program need to feel a sense of shared ownership, particularly as school staff and lived experience worker perspectives were not collected. Please clarify.

Recommendations, Page 27, Line 611: Do you mean lived experience organisation here (not patient organisation)?

Recommendations, Page 26-27: Do you have any recommendations for assessing whether these strategies are successful in improving program implementation?

Conclusions, Page 27, Line 625: Do you mean effectiveness of the preparation phase, rather than efficiency?

6. PLOS authors have the option to publish the peer review history of their article (what does this mean?). If published, this will include your full peer review and any attached files.

Reviewer #1: No

Reviewer #2: No

---

## [Author Response · Author response to Decision Letter 0]

20 Aug 2021

Many thanks for the elaborate comments of both reviewers. We appreciate the time and effort both reviewers took to review our manuscript. All of these comments helped to improve the manuscript. 

The comments of reviewers 1 and 2 are numbered; followed by our responses, while changes made in the manuscript are between quotation marks. 

We have numbered the reviewer comments. Certain replies include references to revisions beneath another reviewer comment as to reduce the wordiness of this document. 

The lines and page numbers correspond to the marked version of the revised manuscript. Thank you for your consideration of this revised manuscript. 

On behalf of all authors,

Mandy Gijzen, MSc. 

Reviewer #1: 

This study presents a process evaluation of a large-scale study of a serious game delivered in secondary schools by a lived experience worker. The inclusion of people with lived experience in the delivery of the program is positive; however, a critical problem with this study, is that it states that it intends to do a process evaluation, but it appears to not be measuring the rollout of the program against a specific process to assess how effective it was. The study presents more as a feasibility study, but this is also problematic as not all groups were included as participants so that the true feasibility of the program can be appropriately evaluated e.g., the discussion makes some important points about how to recruit and support lived experience workers, but this information does not seem to have been gained through the current study results This is because the lived experience workers were not asked about their involvement in the program, e.g., what was difficult for them, and what could have helped – instead the authors rely on assumptions. This seems like a large oversight for the conclusions the authors wished to reach, and is also noted as a limitation – “Furthermore, the current study did not directly tap into the experiences of the students, the LEW, teachers and school administrators on how they experienced the Moving Stories program”. I think perhaps a reframing of the study would help, as I don’t believe that currently this study supports the “feasibility” of the program as stated in the conclusions because it did not tap into those critical experiences.

My specific comments to the authors are below:

We agree, our study is better described as a feasibility study, not as a process evaluation. We changed the subtitle of the MS accordingly. The subtitle no longer reads “Process Evaluation”, it now reads “Feasibility of scaled-up implementation”

Abstract

1. The abstract should be more specific about what was done and what was found – do not list what you will present in the paper later on, this information should be presented here as well. At the moment, the methods includes a description of the Moving Stories program, but not what was actually done – were there any participants, list these here, how was the data for the results collected? Also, more information should be given in the results – it says that “28.9% of trainees were able to carry out their training” – how many trainees were there overall?

We re-wrote the abstract to better clarify the aims of the study and to include more information regarding participants, data collection and outcomes.

Changes in manuscript:

Background: Stigma and limited mental health literacy impede adolescents getting the help they need for depressive symptoms. A serious game coupled with a classroom session led by lived experience workers (LEWs) might help to overcome these barriers. The school-based Strong Teens and Resilient Minds (STORM) preventive program employed this strategy and offered a serious game, Moving Stories. The current study was carried out to assess the feasibility of scaled-up implementation of Moving Stories within the STORM program. 

Methods: Moving Stories was offered in three steps: (1) introductory classroom session, (2) students playing the game for five days, (3) debriefing classroom session led by lived experience worker. Data was collected on the number of participating students, costs of offering Moving Stories, and was further based on the notes of the debriefing sessions to check if mental health first aid (MHFA) strategies were addressed.

Results: Moving Stories was offered in seven high-schools. Coverage was moderate with 982 participating students out of 1880 (52%). Most participating students (83%) played the Moving Stories app three out of the five days. Qualitative data showed that the MHFAs were discussed in all debriefing sessions. Students showed great interest in lived experience workers’ stories and shared their own experiences with depression.

Conclusions: Implementation of Moving Stories to prevent depression in high-school students is feasible, but not without some struggles. Future implementation and scale-up of Moving Stories could benefit from improved selection and training of LEWs that played such an important role in grabbing the full attention of students and were able to launch frank discussions about depressive disorder and stigma in classrooms.

Introduction:

2. “Depression symptoms, if ignored, are likely to develop into major depression disorder [1]” I believe this article is trying to demonstrate that the experience of major depression in childhood, increases the risk of depression in adulthood; thus, I would suggest revising the sentence.

This sentence has been deleted as part of an extensive language review suggested by reviewer 2 to reduce wordiness.

Methods:

3. Participants: The age range seems very broad (11-15 years) for only 8th grade students – can you please clarify why this was the case?

We look at 8th grade students who usually fall into the age bracket of 13-14 years. However, as we did not want to exclude any 8th grade students the age range was expanded to 11-15 years old. We have added this information in our manuscript.

Changes in manuscript:

Methods, page 9, lines 183-186:

“During their second year of high school most student in The Netherlands are 13-14 years of age. However, there are few exceptions and we did not want to exclude any students. Therefore, a wide age range of 11-15 years was used for inclusion.”

4. Important to confirm if the ethics statement listed at the beginning in the table for the journal will be included in the final article, as otherwise I would list this here.

An ethics statement has been included in the Methods section.

Changes in manuscript:

Methods, page 14, lines 311-316:

“The medical ethics committee (Commissie Mensgebonden Onderzoek Region Arnhem-Nijmegen) in The Netherlands approved this study (NL61599.091.17). Informed consent from adolescents and their parents was obtained for the STORM trial. During the second trial year participation in Moving Stories was allowed on voluntary basis for students not participating in the STORM trial. At all times, students were informed that a researcher would be present during the debriefing sessions who might take anonymous notes.”

5. Procedure: Could you explain in more detail how the program was delivered? How many schools were involved, how many teachers, how many lived experience workers? This sort of information seems important for a process evaluation, it should be included in the methods (not just the results) and should also be included in the abstract where indicated above.

The Participants and Procedure section has been expanded to present all relevant information. Moreover, we have added a specific paragraph regarding the LEW. 

Changes in manuscript:

Methods, pages 8-9, line 169-180:

“In the context of the STORM trial, the program of Moving Stories was implemented in two consecutive study years, namely 2017-2018 and 2018-2019. In the Netherlands the study year generally starts around September and concludes around July. Moving Stories was offered from January to May in each study year. Fifteen high schools participate within the STORM trial, of which seven were randomly assigned to the experimental condition. Moving Stories was only offered at schools in the experimental condition.

Nearly one thousand (N = 982) eight grade high-school students across seven schools in the Southeast of the province North-Brabant participated in Moving Stories. Data from the STORM trial indicated that 11-13% of these eight grade students were screen positive for depressive symptoms as measured by the Children’s Depression Inventory 2, and 5-6% of them were screen positive on suicidal thoughts or behaviors.”

Methods, page 13, line 285:

“Any LEW that applied was invited to participate in the training (n = 37).”

6. Did students have access to the program from January to May? Or just over the five days? If this could be described a bit more clearly that would be helpful.

Students had access to the program for the five days. After five days they could no longer actively perform any actions within the game. This has been clarified in the Methods section.

Changes in manuscript:

Methods, pages 10-11, lines 224-228:

“Students then go on to play the game app for 10-15 minutes in the morning, between 7AM and 11AM, for 5 days. Students only had access to the app during the five days that they were asked to play, while they could still open the app students could no longer perform any actions or start over after the five playing days of their class was over.”

7. What specific data analyses were conducted? List these here under a heading: Data analysis.

We have added a new heading “Data analysis” as suggested by the reviewer. 

Changes in manuscript:

Methods, page 16, lines 350-360:

“Descriptive statistics such as means and standard deviations were computed in SPSS Version 27 [33]. With regards to the qualitative data in the form of notes, first we checked in how many sessions the required five aid strategies were discussed by the LEW. Additionally, we checked whether any common themes arose across the different debriefing classroom session. These themes were not pre-determined as there was no specific theme the LEW had to discuss with regard to their experiences or depressive symptoms in general. Instead, they were free to discuss several aspects of both the game and their personal experience with depressive symptoms and were encouraged to interactively tailor the session to the interests expressed by the students. Discussed topics were considered a theme if a topic occurred in at least two of recorded notes. A researcher read and coded all interviews using MAXQDA 2020 Version 2.4.0 [34]. Quotes representative for themes were then exported into Excel.”

Results

8. The description of the training required (8 hrs for LEWs) should be included in the method, and then the cost analysis of this can be the results.

Information regarding the training has been relocated to the Methods section under a new subheading: Training of Lived Experience Workers.

Changes in manuscript:

Methods, pages 13-14, lines 278-309:

“The LEW were recruited via a post on the Facebook page of the Trimbos Institute (Netherlands Institute of Mental Health and Addiction) and GGZ Oost Brabant (Mental Health Service). Additionally, we contacted several lived experience organizations. Requirements were that the LEW had some prior lived experience with depression. LEWs were informed during the recruitment process that they were required to tell their personal story to students in their second year of high school and that they would need to complete a training beforehand. Any LEW that applied was invited to participate in the training (n = 37).

The LEWs first had to complete an 8 hr training. This training for LEWs was designed by the authors, and a researcher who is also a LEW and was an experienced trainer. The training sessions were provided by these researchers as well. At the start of the training, participants were asked to play the serious game to fully understand what the students have done prior to their debriefing session.

The training itself consisted of a theoretical part and a practical part. During the theoretical part (3hrs) the LEWs learned more about the STORM program, its delivery and the goals of the Moving Stories intervention, and what their role was in the Moving Stories intervention. In the practical training (5hrs), the LEWs had the opportunity to practice telling their personal story in 10-15 minutes, and introducing the five MHFA strategies. The discussion of the five strategies from the MHFA was the most important component of the debriefing session and the LEWs were required to discuss all five strategies. Additionally, LEWs had to prepare themselves for questions that students might ask regarding their experiences, difficult topics (such as students having suicidal thoughts) and the game. 

 The LEW were provided with training materials (PowerPoints used during the training session) and a PowerPoint that they were free to use or adapt to their liking for the debriefing session. This PowerPoint consisted of a few example questions that the LEW could ask students as well as a slide with the five strategies from the MHFA. 

The training sessions took place from September to December in both years indicating a time lapse of one to six months between training and the debriefing session. In the meantime the LEWs were updated on planning by the researchers via email and WhatsApp, but any other contact between the researchers and LEWs was minimal during the training and debriefing sessions. By design, LEWs always led the debriefing sessions in the presence of a gate keeper-trained teacher and a researcher, with the researcher taking notes.”

9. Again, the information listed here about the number of schools etc., needs to be in the methods. The results can remain, but it is important to list who was involved (participants) and what you did (procedure) in the methods first.

Information regarding number of schools and participants have been added in the Methods section as can be found underneath the changes made for reviewer comment number 5.

10. How were the themes for the ‘debriefing sessions” identified - if this qualitative analyses, this needs to be explained and listed in the method? Also what was the purpose of analysing this information – was it to use it to improve the program? Or just out of interest?

Information regarding themes and analyses have been added in a new subsection: Data analysis. The changes in the manuscript can be found underneath the changes made for reviewer comment 7.

Discussion

11. Could you give an overview of what was found under the principal findings heading, it would be helpful to really identify what the key findings were and list them here.

We have added a paragraph with main findings under the Principal Findings heading as suggested.

Changes in manuscript:

Discussion, pages 27-28, lines 618-630:

“Among the 1880 eligible students 982 (52%) downloaded the Moving Stories app and 83% of them played the game for at least one day with an average of three days out of the five days. The LEWs discussed the five MHFA strategies with students in all recorded sessions. The notes of the debriefing sessions indicated that sadness-like symptoms were readily recognized as indicative of depression, whereas dismissive or grumpy behaviors were not. The students showed great interest in the personal stories of the LEWs and in more than half of the debriefing sessions one or more students shared their own personal experiences with mental health and depressive symptoms. This may suggest that mental health literacy was enhanced and stigma reduced among students. The total cost of implementing Moving Stories across seven high schools was €11,043 (US$ 12,367), of which €4,672 (US$ 5,232) was spent on the training of the LEWs. However only 11 out of the 37 trained LEWs (30%) carried out their training in practice and led a total of 49 debriefing sessions. This suggests that selection and training of LEWs should be improved.”

12. There are some critical statements that are made in the discussion, which seem to overstep the findings of the current study. For example, it is not clear where the data that inform the statement “it is important that each person who has a role in the program feels involved in the program and is motivated to bring the program to success” – or “it is important that each person is aware of the program components” how do you know this, what data was uncovered in your study that tells you this is the case? There is very little in regard to information collected from either the schools or the LEWs, so it is difficult to see where these statements have come from.

Our discussion has been streamlined to reflect only the findings that are mentioned in the Results section. As such we have deleted several paragraphs rearranged the order of certain paragraphs and revised our text.

Changes in manuscript:

Discussion, preparing Moving Stories, pages 29-31, lines 651-717:

“Recruitment and training of 37 LEWs costed €4,672 ($5,232) or €126 ($141) per LEW. The total costs (including recruitment and training costs of LEWs) of offering Moving Stories to 7 high schools was €11,034 ($12,357) or €1,576 ($1,765) per school. Since 982 students participated in Moving Stories, this amounted to €11 ($13) per participating student. This was achieved below the planned budget ceiling. However, it should be noted that of the 37 trained LEWs, only 11 led debriefing sessions, which suggests that recruitment and training of the LEWs should be improved.

For this project, every LEW who indicated they wanted to be involved in the project, was allowed to complete the training. Thus, it would be beneficial to develop a more specific set of criteria for selecting a LEW and make it very clear what is expected of LEW. Selection criteria that we would recommend are that the LEW has some prior experience in delivering their personal story to a public, preferably with high school students in classrooms. Recruitment of LEWs via LEW organizations could help finding qualified LEWs.

The high drop-out rate among the LEWs further suggests the need to more intensively involve the LEWs in the design of the debriefing session and how it is implemented. After all, their engagement and sense of ownership may hold the key to their willingness to turn the debriefing sessions into a success. After all, a sense of belongingness is important [35]. In addition, the training session for the LEWs may need to be better geared towards acquiring the ability and confidence to provide debriefing sessions in the classroom setting [36]. We refer to research by Chen et al. [37] who identified four main activities that are needed to adequately prepare a LEW for contact-based interventions. First, LEWs need to be psychologically prepared: they need adequate support and confidence. Second, adequate knowledge and skills need to be built. Third, a LEW needs to develop their personal story in a way they can share it. Fourth, and most importantly, the LEWs need to be able to practice the entire session.

 Therefore, we recommend more time is allocated to the practical part of the training such that each LEW has ample training time to develop and tell their personal story in 10-15 minutes , and where the LEW can practice how to introduce the five MHFA strategies and learns how to anticipate to the various reactions and questions that students might bring up in real debriefing sessions. As a case in point, a prior study found that a transient dip in one’s mood could occur within hours or first couple of days after having shared one’s personal experience related to mental – although in the long run one’s mental health will not be affected and LEWs may feel that sharing their story was beneficial to themselves and others [35]. 

 The high drop-out rate among the LEWs may also indicate that the level of contact between the research staff and the LEWs in the period between training and the debriefing sessions was too low. This suggests another area where improvement can be made. Unfortunately, the current study did not ask the LEWs how the level of contact with the researchers affected their engagement in the implementation of Moving Stories. For future evaluations, it is important that the experiences of LEW are evaluated to determine which aspects or processes might impede further participation especially during the implementation.” 

13. “A more efficient method of selecting the LEW needs to be developed before LEW are offered a training as part of their involvement in the Moving Stories intervention” – what does this mean and how did you come to this conclusion from your data? Did you ask LEWs about their involvement and any barriers they faced? There is a line in the results that says that some indicated that it interfered with their work, but it is not clear if this information was collected as data for the current project (e.g., were LEWs participants in the study too?).

We have added more information regarding the recruitment of the LEW in the Methods section. The changes in the manuscript can be found underneath the changes made for reviewer comment 8.

Additionally, we have revised the section in the Discussion regarding the training to reflect the findings from the Methods and Results. The changes in the manuscript for the Discussion can be found underneath the changes made for reviewer comment 12.

Conclusion

14. I am not sure that you can say the feasibility of the program has been demonstrated via this study. If anything, there seemed to be issues with the LEWs and their participation and there is no data collected from the schools about how it fit into the curriculum etc., so it is unclear how this statement was reached based on the data presented.

We have rewritten our conclusion to more adequately reflect the revised version of the Discussion.

Changes in manuscript:

Conclusion, page 37, lines 857-863:

“Concluding, implementation of Moving Stories to prevent depressive symptoms in high-school students is feasible. However, there were few trained LEWs that were able to provide debriefing sessions suggesting that selection and training of LEWs needs to be improved. The role of the LEWs is important because they managed to grab the full attention of students and were able to launch frank discussions about depression and create a space wherein students were willing to share their personal experiences. This suggests that Moving Stories was successful in enhancing mental health literacy and reducing depression stigma.”

 

Reviewer #2: 

Manuscript title: Implementing a serious game led by lived experience workers for depression prevention in high-school students: process evaluation

General comments: This manuscript describes a process evaluation of a high-school-based universal depression prevention program, which combines a serious smartphone game and an in-person discussion session with a lived experience worker. Some interesting reflections are provided on the challenges of training and maintaining lived experience workers, and the successes of generating classroom discussions through a debriefing session delivered by a lived experience worker. However, greater clarity around the objectives, methods and interpretation of findings is needed. The manuscript may also benefit from incorporating some discussion of how the lessons learned from implementing Moving Stories can be applied to other similar programs. The manuscript is generally clearly written. While I have noted some points of ambiguity in my specific comments, I do recommend the authors review the manuscript for any typographical or grammatical errors that are not specifically noted during review. I have made a number of specific comments below, I hope they are helpful to the authors in revising the manuscript.

We have reviewed the entire manuscript for any typographical or grammatical errors as well as to reduce wordiness and possible ambiguity. To reduce the wordiness of this reply, we have not pasted all changes made in this regard. They have been marked using track changes in the revised manuscript. 

Specific comments:

1. Title and Abstract: Moving Stories is characterised as a serious game led by lived experience workers. This seems misleading, as the only lived experience involvement described in the manuscript is the role of the lived experience workers in delivering a debriefing session. This element of the program was delivered by people with lived experience, but there is no evidence of true lived experience leadership. Please clarify whether this program involved lived experience leadership (i.e., the game was developed by people with lived experience, the program was developed and run by people with lived experience). If this is not the case, please adjust the title and abstract to accurately represent the nature of the Moving Stories program.

We have adjusted the title and abstract to accurately reflect that only the classroom debriefing session was led by lived experience workers. 

Changes in manuscript:

Title, page 1:

“Serious game coupled with a contact-based session led by lived experience workers for depression prevention in high-school students: feasibility of scaled-up implementation.” 

Abstract, page 3, lines 35-38:

“A serious game coupled with a classroom session led by lived experience workers (LEWs) might help to overcome these barriers.”

Introduction:

2. Page 6, Lines 109-110: What is meant by “a game-based design was used for Moving Stories led by lived experience workers.” Was the design and implementation of the program led by lived experience workers, or does this refer to the delivery of the briefing session? Please ensure the level of lived experience involvement is accurately characterised.

We agree that way the level of lived experience involvement could have been reflected on more accurately. We made the following changes in the manuscript.

Changes in manuscript:

Introduction, Page 7, lines 129-131:

“Therefore, Moving Stories was based in both approaches: education as well as direct-contact with a lived experience worker.”

3. Page 6, Lines 117-118: What were the specific objectives of the process evaluation? What elements of program implementation did you plan to evaluate? Outlining these objectives in the Introduction section would provide more context for the methods and findings presented below. At the moment it is somewhat unclear why certain kinds of data were collected (e.g., costs) and others were not (e.g., perspectives of lived experience workers, school staff, and students on acceptability/feasibility).

The specific objectives of the process evaluation have been added as suggested by the reviewer.

Changes in manuscript:

Introduction, page 7-8, lines 150-162:

“Therefore, the primary aim of the current study was to assess the feasibility of implementing Moving Stories as a universal preventive intervention within the STORM program. More specifically, we wanted to know (1) the number and percentage of high-school students that actively played Moving Stories (coverage rate), (2) the average number of playing days out of the five (participation rate), (3) how successful LEWs were in reviewing all five mental health first aid (MHFA) strategies during the debriefing sessions and to what extend students could identify depressive symptoms (fidelity), (4) what other themes were addressed and resonated with the students during the debriefing sessions, and how the students responded to the personal experiences of the LEWs. Finally, implementation costs were computed including the costs of recruiting and training LEWs for leading the debriefing sessions to see if money was spent wisely or requires reallocation. As recommended by Oakley et al. [31], assessing the feasibility of implementing an intervention is performed before the effectiveness data of the STORM trial are analyzed.”

Materials and Methods

4. Participants and Procedure, Page 6: How many schools were involved in the trial? Were different types of schools involved (e.g., government vs private funded)? What level of diversity did schools represent (e.g., average socioeconomic status of student population, range of geographical areas)?

Information regarding schools has been added in the Methods section. In addition, level of education of schools was added in the Results section in Table 1. 

Changes in manuscript:

Methods, Participants and procedure, page 8-9, lines 169-180: 

“In the context of the STORM trial, the program of Moving Stories was implemented in two consecutive study years, namely 2017-2018 and 2018-2019. In the Netherlands the study year generally starts around September and concludes around July. Moving Stories was offered from January to May in each study year. Fifteen high schools participate within the STORM trial, of which seven were randomly assigned to the experimental condition. Moving Stories was only offered at schools in the experimental condition.

 Nearly one thousand (N = 982) eight grade high-school students across seven schools in the Southeast of the province North-Brabant participated in Moving Stories. Data from the STORM trial indicated that 11-13% of these eight grade students were screen positive for depressive symptoms as measured by the Children’s Depression Inventory 2, and 5-6% of them were screen positive on suicidal thoughts or behaviors.”

5. Participants and Procedure, Page 7: How were students recruited? How did they provide informed consent (for app and research)? Were students required to participate in the research in order to access the Moving Stories program and participate in the debriefing session?

Information regarding consent have been added in the Data collection section. 

Changes in manuscript:

Methods, Data collection, page 14, lines 311-316:

“The medical ethics committee (Commissie Mensgebonden Onderzoek Region Arnhem-Nijmegen) in The Netherlands approved this study (NL61599.091.17). Informed consent from adolescents and their parents was obtained for the STORM trial. During the second trial year participation in Moving Stories was allowed on voluntary basis for students not participating in the STORM trial. At all times, students were informed that a researcher would be present during the debriefing sessions who might take anonymous notes.”

6. Intervention description, Page 7, Line 148: “Lived Experience Worker of depression” doesn’t quite make sense. I’d suggest changing this to “Lived Experience Worker with experience of depression” (or something similar) throughout the manuscript.

We have changed the term “lived experience worker of depression” to “lived experience worker with experience of depression” throughout the manuscript as suggested.

7. Intervention description, Page 7, Lines 149-150: This section should specify that the introductory lesson was led by a researcher. Additionally, please specify what information students were given in the introductory session (i.e., what were they told the aim of the program was? Is depression specifically mentioned?).

The intervention description of the introductory session has been expanded.

Changes in manuscript:

Methods, Intervention, page 10, lines 210-222:

“During the introductory lesson a researcher would explain to students how to download the app, what was expected of them and provided log-in codes. These sessions were delivered during a regular school class in a structured manner using a ready-to-use PowerPoint presentation to ensure every student received similar instructions. The aim of the game, as described to students, is that they will enter a virtual home where they will find a girl, Lisa, who is not feeling well. Students are asked to see if they can be of any help to Lisa. The term depression is not used during this introductory session. Although the game is played individually, the students are encouraged to help each other in identifying strategies that could be of any help to Lisa. Students were instructed to download the app during this introductory classroom session. After downloading the app, participants had to enter a passcode in order to receive access to the full app to ensure only adolescents from participating schools could actively use the app. As such, only participants that were enrolled in their second year of high school could participate.”

8. Intervention description, Pages 7-8, Lines 151-155: Some elements of program delivery need clarification. When did students complete the game – in class or in their own time? Were they prompted to play the game each day, if so how? Why could the game only be played for a limited time period in the morning, particularly if the game was played outside of class?

The intervention description of playing the game has been expanded.

Changes in manuscript:

Methods, Intervention, page 10-11, lines 224-244: 

“Students then go on to play the game app for 10-15 minutes in the morning, between 7AM and 11AM, for 5 days. Students only had access to the app during the five days that they were asked to play, while they could still open the app students could no longer perform any actions or start over after the five playing days of their class was over. Students play the game individually, but simultaneous with their classmates and they are expected to do this in their own time. Schools did not provide the students with time to complete the game during school hours. In the game students learn to interact with a virtual girl, Lisa, suffering from depressive symptoms. The students perform actions in the game (over 5 days) which consist of household tasks and/or offering Lisa help, and have the ability to talk to Lisa in a structured manner using preprogrammed options conversation). The aim is that they learn effective strategies when helping someone with depressive symptoms or to become more able to seek help for themselves. Students were prompted to play the game as reminders from the app would pop-up when Lisa’s house would open (7AM) and an hour before the house would close (10AM). As with any mobile app, students had the ability to turn of pop-ups from the app. Feedback is provided each day by gaining or losing points indicating a relationship score indicating the quality of the relationship with Lisa based on the action they did in the game. In addition, they receive written text messages from Lisa telling the students how Lisa felt about each action they performed. In this way, students knew why they had gained or lost points. These messages were sent after school hours, from 3PM until 7PM.”

9. Intervention description, Page 7, Lines 160-163: How long was the debriefing session? Aside from the requirement to cover the five MHFA strategies, were there any other guidelines or requirements for the structure of the session?

The intervention description of the debriefing session has been expanded.

Changes in manuscript:

Methods, Intervention, page 11-12, lines 245-266:

“After approximately seven days, students are brought back together again for the debriefing session. This debriefing session is an in-person classroom session lead by the LEW during a regular school class which are between 40 and 50 minutes long depending on the school. The LEW was aware of the length of each sessions beforehand. The LEW talks about his or her personal experience (approximately 10 minutes) and relates these personal experiences with depression to the game, which is then related to the students’ experiences with the game. The LEWs are required to discuss the five first aid strategies from the teen Mental Health First Aid (MHFA): (1) look for warning signs, (2) ask how their friend or peer is doing, (3) listen without judgement, (4) help them connect with an adult, and (5) your friendship is important [21]. The five strategies are reviewed and wrapped up as take-home messages for the students during the debriefing sessions. For this the LEWs can choose to use the PowerPoint slides that were made available to them during their training. In the PowerPoint slides show some example questions to launch discussions about the five aid strategies, but the LEWs were free to use these slides and adapt it to their personal preference as long as the five strategies are discussed. Moreover, the structure of the debriefing session was deliberately left open so LEWs could adapt it to their personal preference and make impromptu responses to the themes brought up by the students.”

10. Data collection, Page 9, Line 183: Information about the cost of the Moving Stories app was not reported. Please add this to the Results or remove this statement from the Methods.

This statement has been removed from the Methods section.

11. Data collection Page 9, Line 188: Costs associated with debriefing sessions are already listed in the paragraph above, statement can be removed from this paragraph.

The statement regarding “costs associated with debriefing session” have been removed.

12. Data collection, Page 9, Lines 191-193: Was informed consent obtained from students and lived experience workers permitting the collection and reporting of session minutes? This is particularly important as individual students and workers have been quoted in the Results section.

Information regarding consent have been added in the Data collection section. The revisions can be found under the changes made underneath reviewer comment 5. 

13. Data collection, Page 9, Lines 191-193: The spread of recorded sessions across schools should be noted here.

We have added further information regarding the recorded sessions.

Changes in manuscript:

Methods, Data collection, page 15, lines 332-335: 

“ In total, 49 debriefing sessions were provided by 11 LEWs. Notes were taken of 18 debriefing sessions: 8 during study year of 2017-2018 and 10 during 2018-2019 with at least one debriefing session per school (see Table 1). These notes were kept by four researchers to record what the LEW and students discussed during the debriefing sessions.”

14. Data collection, Page 9, Lines 194-199: Please describe the analysis methods used to produce themes from the debriefing session minutes.

We have added a section on Data Analysis.

Changes in manuscript:

Methods, Data analysis, page 16, lines 350-360:

“Descriptive statistics such as means and standard deviations were computed in SPSS Version 27 [33]. With regards to the qualitative data from the notes, first we checked in how many debriefing sessions the required five aid strategies were discussed by the LEW. Additionally, we checked whether common themes arose across the different debriefing sessions. These themes were not pre-determined as there was no specific theme the LEW had to discuss with regard to their personal experiences or depressive symptoms in general. Instead, they were free to discuss several aspects of both the game and their personal experience with depressive symptoms and were encouraged to interactively tailor the session to the interests expressed by the students. Discussed topics were considered a theme if a topic occurred in at least two of the recorded notes. A researcher read and coded all notes using MAXQDA 2020 Version 2.4.0 [34]. Quotes representative for themes were then exported into Excel.”

Results:

15. Preparing delivery, Page 10: How were Lived Experience Workers recruited and selected? Were they required to have any previous experience? How was the training program developed? Was the training program developed with any lived experience input, or input from organisations with experience in delivering lived experience content in classroom settings? Who ran the training sessions? Were any materials provided to training participants for future reference/revision? Given the focus of the Discussion and Conclusion on the need to improve training and support of Lived Experience Workers, it is important to provide these details.

We have added more information regarding the selection and recruitment of the LEW in the Methods section in separate section.

Changes in manuscript:

Methods, Recruitment and Training of Lived Experience Workers, page 13-14, lines 279-309:

“The LEW were recruited via a post on the Facebook page of the Trimbos Institute (Netherlands Institute of Mental Health and Addiction) and GGZ Oost Brabant (Mental Health Service). Additionally, we contacted several lived experience organizations. Requirements were that the LEW had some prior lived experience with depression. LEWs were informed during the recruitment process that they were required to tell their personal story to students in their second year of high school and that they would need to complete a training beforehand. Any LEW that applied was invited to participate in the training (n = 37).

The LEWs first had to complete an 8 hr training. This training for LEWs was designed by the authors, and a researcher who is also a LEW and was an experienced trainer. The training sessions were provided by these researchers as well. At the start of the training, participants were asked to play the serious game to fully understand what the students have done prior to their debriefing session.

The training itself consisted of a theoretical part and a practical part. During the theoretical part (3hrs) the LEWs learned more about the STORM program, its delivery and the goals of the Moving Stories intervention, and what their role was in the Moving Stories intervention. In the practical training (5hrs), the LEWs had the opportunity to practice telling their personal story in 10-15 minutes, and introducing the five MHFA strategies. The discussion of the five strategies from the MHFA was the most important component of the debriefing session and the LEWs were required to discuss all five strategies. Additionally, LEWs had to prepare themselves for questions that students might ask regarding their experiences, difficult topics (such as students having suicidal thoughts) and the game. 

 The LEW were provided with training materials (PowerPoints used during the training session) and a PowerPoint that they were free to use or adapt to their liking for the debriefing session. This PowerPoint consisted of a few example questions that the LEW could ask students as well as a slide with the five strategies from the MHFA. 

The training sessions took place from September to December in both years indicating a time lapse of one to six months between training and the debriefing session. In the meantime the LEWs were updated on planning by the researchers via email and WhatsApp, but any other contact between the researchers and LEWs was minimal during the training and debriefing sessions. By design, LEWs always led the debriefing sessions in the presence of a gate keeper-trained teacher and a researcher, with the researcher taking notes.”

16. Preparing delivery, Page 10: The utility of reporting costs is unclear. Was a cost-benefit analysis conducted? How do these costs compare to other programs? From the information presented, it is not possible for the reader to determine whether the program represented good value for money.

We have added in the Introduction a more specific aim of why we reported the costs. Moreover, in the Discussion we have added a section on costs. 

Changes in manuscript:

Introduction, page 8, lines 158-160:

“Finally, implementation costs were computed including the costs of recruiting and training LEWs for leading the debriefing sessions to see if money was spent wisely or requires reallocation.”

Discussion, page 29, lines 651-657:

“Recruitment and training of 37 LEWs costed €4,672 ($5,232) or €126 ($141) per LEW. The total costs (including recruitment and training costs of LEWs) of offering Moving Stories to 7 high schools was €11,034 ($12,357) or €1,576 ($1,765) per school. Since 982 students participated in Moving Stories, this amounted to €11 ($13) per participating student. This was achieved below the planned budget ceiling. However, it should be noted that of the 37 trained LEWs, only 11 led debriefing sessions, which suggests that recruitment and training of the LEWs should be improved.”

17. Offering Moving Stories, Pages 10-11: Again, the utility of reporting costs is unclear. If costs are to be reported, perhaps summarising costs across preparation and implementation in a single table would clarify the total cost of running the program?

We have added an overview of the costs in a table as suggested.

Changes in manuscript:

Results, Offering Moving Stories, page 20, lines 433-434:

Table 2. Overview of cost in euros.

 Costs Total Costs % of total costs

Preparation 4,672 42%

Trainer 3,798 

LEW 874 

Offering 6,371 58%

Materials 1,750 

LEW 4,621 

Total Costs 11,043 

18. Offering Moving Stories, Page 12, Lines 250-259: Is any information available about how Lived Experience Workers were communicated with and kept engaged with the program between training and classroom sessions? This may help shed some light on possible areas for improvement when next implementing the program.

We have added information regarding engagement with LEW throughout the intervention program in the “Recruitment and Training” section of the Methods as well as added this in the Discussion.

Changes in manuscript:

Methods, Recruitment and Training, page 14, lines 304-309:

“The training sessions took place from September to December in both years indicating a time lapse of one to six months between training and the debriefing session. In the meantime the LEWs were updated on planning by the researchers via email and WhatsApp, but any other contact between the researchers and LEWs was minimal during the training and debriefing sessions. By design, LEWs always led the debriefing sessions in the presence of a gate keeper-trained teacher and a researcher, with the researcher taking notes.”

Discussion, page 31, lines 711-717:

“The high drop-out rate among the LEWs may also indicate that the level of contact between the research staff and the LEWs in the period between training and the debriefing sessions was too low. This suggests another area where improvement can be made. Unfortunately, the current study did not ask the LEWs how the level of contact with the researchers affected their engagement in the implementation of Moving Stories. For future evaluations, it is important that the experiences of LEW are evaluated to determine which aspects or processes might impede further participation especially during the implementation.”

19. Debriefing sessions, Page 13: Beyond the MHFA topics, was there any guidance on the structure of the debriefing sessions? This may have influenced what was most commonly discussed. It is important to outline the structure of the debriefing sessions in the Methods.

We have expanded our description of the debriefing sessions in the Methods as can be found underneath the changes made for reviewer comment number 9.

20. Debriefing sessions, Page 16, Line 359: Here and in a few other places in the manuscript, school mentors are referred to. What is a school mentor? This term needs defining for an international audience.

The term school mentor has been deleted in most instances and replaced by the term “school teacher”. In the quotes by the students an explanatory note has been added. 

Changes in manuscript:

Results, page 24, lines 527-529:

“Note: A mentor is a teacher who also acts as a trusted guide for students in matters related to their study. Typically, a mentor is assigned at the start of school year to either one or more classes, while a class typically has some 20-30 students.”

Discussion:

21. Program costs are not discussed in the Discussion section. This makes me unsure of the utility of reporting them – what was the purpose of assessing costs? What do the costs mean in practical terms?

We have added a section on the program costs in the Discussion.

Changes in manuscript:

Discussion, Preparing Moving Stories, page 29, lines 651-657:

“Recruitment and training of 37 LEWs costed €4,672 ($5,232) or €126 ($141) per LEW. The total costs (including recruitment and training costs of LEWs) of offering Moving Stories to 7 high schools was €11,034 ($12,357) or €1,576 ($1,765) per school. Since 982 students participated in Moving Stories, this amounted to €11 ($13) per participating student. This was achieved below the planned budget ceiling. However, it should be noted that of the 37 trained LEWs, only 11 led debriefing sessions, which suggests that recruitment and training of the LEWs should be improved.” 

22. Preparing Moving Stories, Pages 20-21, Lines 450-461: While some interesting points are made here, it is unclear how much of this content relates to the findings reported in the Methods section. Please clarify the links between these assertions and your process evaluation data. Additionally, this section feels like it is missing citations – please add in citations to reference materials as appropriate.

We have deleted this section as it was not directly related to the key findings. 

23. Preparing Moving Stories, Pages 21-22: Parts of the discussion around lived experience workers feels like it is focused on perceived weaknesses of the workers themselves (e.g., they may not be psychologically ready to work in a classroom), rather than focusing on elements of the program and its implementation that could be improved to better support a lived experience workforce. I do not think this is intentional, however, I think it would be beneficial to review the discussion points around the lived experience workers to ensure they are focused on evaluation findings about the program, rather than assumptions about the workers. It may also be helpful to highlight the lack of data on Lived Experience Worker experiences in these discussions – it is hard to propose effective support strategies without knowing what workers enjoyed about delivering the program and what they would change.

We have revised our Discussion around the LEW to better reflect the findings reported in the Results.

Changes in manuscript:

Discussion, Preparing Moving Stories, pages 29-31, lines 660-717:

“For this project, every LEW who indicated they wanted to be involved in the project, was allowed to complete the training. Thus, it would be beneficial to develop a more specific set of criteria for selecting a LEW and make it very clear what is expected of LEW. Selection criteria that we would recommend are that the LEW has some prior experience in delivering their personal story to a public, preferably with high school students in classrooms. Recruitment of LEWs via LEW organizations could help finding qualified LEWs.

 The high drop-out rate among the LEWs further suggests the need to more intensively involve the LEWs in the design of the debriefing session and how it is implemented. After all, their engagement and sense of ownership may hold the key to their willingness to turn the debriefing sessions into a success. Ultimately, a sense of belongingness is important [35]. In addition, the training session for the LEWs may need to be better geared towards acquiring the ability and confidence to provide debriefing sessions in the classroom setting [36]. We refer to research by Chen et al. [37] who identified four main activities that are needed to adequately prepare a LEW for contact-based interventions. First, LEWs need to be psychologically prepared: they need adequate support and confidence. Second, adequate knowledge and skills need to be built. Third, a LEW needs to develop their personal story in a way they can share it. Fourth, and most importantly, the LEWs need to be able to practice the entire session.

 Therefore, we recommend more time is allocated to the practical part of the training such that each LEW has ample training time to develop and tell their personal story in 10-15 minutes , and where the LEW can practice how to introduce the five MHFA strategies and learns how to anticipate to the various reactions and questions that students might bring up in real debriefing sessions. As a case in point, a prior study found that a transient dip in one’s mood could occur within hours or first couple of days after having shared one’s personal experience related to mental – although in the long run one’s mental health will not be affected and LEWs may feel that sharing their story was beneficial to themselves and others [35]. 

 The high drop-out rate among the LEWs may also indicate that the level of contact between the research staff and the LEWs in the period between training and the debriefing sessions was too low. This suggests another area where improvement can be made. Unfortunately, the current study did not ask the LEWs how the level of contact with the researchers affected their engagement in the implementation of Moving Stories. For future evaluations, it is important that the experiences of LEW are evaluated to determine which aspects or processes might impede further participation especially during the implementation.”

24. Preparing Moving Stories, Page 21, Line 465: Is efficient the best word to describe the improved recruitment strategy? Is ‘specific’ or ‘selective’ a more accurate term for what is meant here?

In accordance with the reviewer's suggestion we have replaced the term “efficient” with “specific”. 

Changes in manuscript:

Discussion, Preparing Moving Stories, page 29, lines 661-663:

“Thus, it would be beneficial to develop a more specific set of criteria for selecting a LEW and make it very clear what is expected of LEW.”

25. Preparing Moving Stories, Page 21, Lines 482-484: Was this a limitation of the implementation of recruitment and training – was the Lived Experience Worker role and what it involved inadequately explained before people volunteered for training?

We have added a section on recruitment and training of LEWs in Methods section. Additionally, we have revised our discussion regarding the LEWs. The changes made in the discussion can be found underneath the changes made for reviewer comment number 23.

Changes in manuscript:

Methods, Recruitment and training of LEWs, page 13, lines 279-285:

“The LEW were recruited via a post on the Facebook page of the Trimbos Institute (Netherlands Institute of Mental Health and Addiction) and GGZ Oost Brabant (Mental Health Service). Additionally, we contacted several lived experience organizations. Requirements were that the LEW had some prior lived experience with depression. LEWs were informed during the recruitment process that they were required to tell their personal story to students in their second year of high school and that they would need to complete a training beforehand. Any LEW that applied was invited to participate in the training (n = 37).”

26. Preparing Moving Stories, Page 22, Lines 487-496: This seems like a relevant place to discuss costs, if they are to be better incorporated into the manuscript.

A discussion regarding the costs have been added to this paragraph. The revisions can be found underneath the changes made for reviewer comment number 21.

27. Offering Moving Stories, Page 22, Lines 502-503: Please note this in the Methods section.

This has been added in the Methods section.

Changes in manuscript:

Methods, page 11, lines 228-230:

“Students play the game individually, but simultaneous with their classmates and they are expected to do this in their own time. Schools did not provide the students with time to complete the game during school hours.”

28. Offering Moving Stories, Page 22, Lines 506-509: Please note this context around difference in app access and encouragement of students to help each other play in the Methods section.

This has been added in the Methods section.

Changes in manuscript:

Methods, Intervention, page 10, lines 216-218:

“Although the game is played individually, the students are encouraged to help each other in identifying strategies that could be of any help to Lisa.”

Methods, Data collection, page 14, lines 313-316:

“During the second trial year participation in Moving Stories was allowed on voluntary basis for students not participating in the STORM trial. At all times, students were informed that a researcher would be present during the debriefing sessions who might take anonymous notes.”

29. Offering Moving Stories, Page 23-24, Lines 529-538: Parts of this paragraph are covered in the Preparing Moving Stories section, perhaps it could be moved up and integrated with other Lived Experience Worker-related topics? Additionally, were there any indications from the evaluation data that a health professional would make Lived Experience Workers feel more comfortable? Or that workers were worried about the impact of delivering the session on their mental health? Making assumptions about the mental state of people who chose not to participate in delivering sessions is potentially stigmatising. As noted above, I recommend refocusing this discussion point to emphasise findings from the evaluation (e.g., students tend to share personal stories during debriefing and may thus benefit from direct access to a health professional, lived experience workers may feel more comfortable working in pairs), rather than assumptions about the Lived Experience Workers.

We have streamlined the paragraphs “Preparing Moving Stories” and “Offering Moving Stories” to both avoid duplication of information as well as focus them more on the specific results of the evaluation to avoid stigmatization of the LEWs. 

The revisions for “Preparing Moving Stories” can be found underneath the changes made for reviewer comments 21 and 23. Any discussion regarding the LEWs is now described in this paragraph and no longer underneath “Offering Moving Stories”. 

30. Debriefing sessions, Page 24, Lines 552-553: Citation for this sentence?

We have added a reference.

Changes in manuscript:

Debriefing Sessions, page 34, lines 774-777:

“This is especially the case for adolescents as these symptoms are considered key symptoms of adolescent depression and not in adult depression [46].”

References, page 42, line 1039-1040:

American Psychiatric Association, American Psychiatric Association. DSM 5. American Psychiatric Association. 2013;70.

31. Debriefing sessions, Page 25, Lines 564-565: Citation for this sentence?

We have added a reference.

Changes in manuscript:

Debriefing Sessions, page 34, lines 789-790:

“Moreover, when students feel comfortable to share their personal experiences with mental health, they are more likely to seek help [47].”

References, page 42, lines 1041-1043:

Corry DA, Leavey G. Adolescent trust and primary care: Help-seeking for emotional and psychological difficulties. Journal of Adolescence. 2017;54:1-8. doi: 10.1016/j.adolescence.2016.11.003

32. Strengths and Limitations, Page 26, Lines 586-588: This information should be noted in the Methods section.

This has been added to the Methods section.

Changes in manuscript:

Methods, page 15, lines 332-335:

“In total, 49 debriefing sessions were provided by 11 LEWs. Notes were taken of 18 debriefing sessions: 8 during study year of 2017-2018 and 10 during 2018-2019 with at least one debriefing session per school (see Table 1). These notes were kept by four researchers to record what the LEW and students discussed during the debriefing sessions.”

33. Recommendations, Page 26, Line 608: While it is an interesting point, I am unsure how the data support the recommendation that partners in the program need to feel a sense of shared ownership, particularly as school staff and lived experience worker perspectives were not collected. Please clarify.

Any statements and reflections regarding a sense of shared ownership have been deleted as they do not adequately reflect in the key findings of the current evaluation. 

34. Recommendations, Page 27, Line 611: Do you mean lived experience organisation here (not patient organisation)?

The recommendations have been integrated with the paragraphs regarding the findings in the Discussion.

35. Recommendations, Page 26-27: Do you have any recommendations for assessing whether these strategies are successful in improving program implementation?

The recommendations have been integrated with the paragraphs regarding the findings in the Discussion. We did adjust the terminology from patient organization to lived experience organization throughout the manuscript. 

36. Conclusions, Page 27, Line 625: Do you mean effectiveness of the preparation phase, rather than efficiency?

We have rewritten the entire Conclusions section to better reflect our revised version of the discussion.

Changes in manuscript:

Conclusion, page 37, lines 857-863:

“Concluding, implementation of Moving Stories to prevent depressive symptoms in high-school students is feasible. However, there were few trained LEWs that were able to provide debriefing sessions suggesting that selection and training of LEWs needs to be improved. The role of the LEWs is important because they managed to grab the full attention of students and were able to launch frank discussions about depression and create a space wherein students were willing to share their personal experiences. This suggests that Moving Stories was successful in enhancing mental health literacy and reducing depression stigma.”

---

## [Decision Letter · Decision Letter 1]

18 Oct 2021

PONE-D-21-17344R1Serious game coupled with a contact-based session led by lived experience workers for depression prevention in high-school students: feasibility of scaled-up implementationPLOS ONE

Geachte Dr. Gijzen,

Thank you for submitting your manuscript to PLOS ONE. After careful consideration, we feel that it has merit but does not fully meet PLOS ONE’s publication criteria as it currently stands. Therefore, we invite you to submit a revised version of the manuscript that addresses the points raised during the review process.

I detail the recommendations in the section below. However, in summary there is a need to respond to some additional concerns. Barring any surprises, I anticipate that this would be able to progress to acceptance for publication but request that these be addressed first. I would anticipate that these wouldn't take long to address, and that I could likely review these without needing to go back to the reviewers for additional comment.

We look forward to receiving your revised manuscript.

Met hartelijke groeten,

Dylan A Mordaunt, MB ChB, FRACP, FAIDH

Academic Editor

PLOS ONE

Journal Requirements:

Additional Editor Comments (if provided):

Thank you for your submission. I took this over from another editor as well as not being able to get opinions from all original reviewers and therefore it has taken a little longer to pull together a decision for your resubmission.

One of the reviewers had provided previous suggestions and was subsequently satisfied with the responses. The other reviewer who was new to this review, has taken the perspective of this being an evaluation rather than an implementation study, and therefore made a recommendation to reject the submission.

I think the comments are important and that the authors should be given an opportunity to respond to them, but as you will see I have not agreed with the recommendation to reject but rather to put these criticisms to the authors. In particular, I put it to the authors to argue which of these warrants addressing pre-publication and which belongs post-publication. PLoS One criteria for publication are by design, a bit different to other journals (https://journals.plos.org/plosone/s/criteria-for-publication), and I am not convinced that the problems identified by the second reviewer are so critical that the manuscript should be rejected. However, I think the reviewers comments were immensely valuable and that the authors should feel challenged to address these adequately for the open review process- that is the peer review record published alongside with the manuscript.

With regards specifically to the journal selection criteria:

1. The study appears to present the results of original research. It appears to be an implementation/scaling of an approach developed in an earlier clinical trial.

2. Results do not appear to have been published elsewhere. The second reviewer has raised the question of "salami slicing" and it would be helpful for the authors to clarify what approach has been taken with regards to the publication of results from the underlying study. My interpretation is that this could be considered an "extension phase" of the original trial, or as the authors have called it, an implementation study focused on the scaling of the capability developed and studies in the original trial. I do think implementation science is a separate issue to clinical trials, but I think clearly distinguishing this is important for the record.

3. Experiments, statistics, and other analyses are largely performed to a high technical standard and are described in sufficient detail. The second reviewer has raised the possibility of whether aspects such as the interviews being recorded with notes rather than with recordings/coding, constitute a critical flaw. In the context of this being an implementation study, I would put it to the authors to respond to this with regards to feasibility in particular. It has been commented on but given that we could anticipate that this might be a criticism post-publication, this may warrant expanding on.

4. Conclusions are presented in an appropriate fashion and are supported by the data.

5. The article is presented in an intelligible fashion and is written in standard English.

6. The research meets all applicable standards for the ethics of experimentation and research integrity.

7. The article adheres to appropriate reporting guidelines and community standards for data availability, however I would suggest that upon clarifying that this is an implementation study rather than an extension phase of a clinical trial, that the authors consider reviewing the manuscript for adherence to the Standards for Reporting Implementation Studies (StaRI) Statement (https://www.equator-network.org/reporting-guidelines/stari-statement/). This is particularly helpful if the manuscript is later included in a systematic review or other form of meta-research, in to maximise the assessed quality of the study (not withstanding any critical problems).

Reviewers' comments:

Reviewer's Responses to Questions

**Comments to the Author**

1. If the authors have adequately addressed your comments raised in a previous round of review and you feel that this manuscript is now acceptable for publication, you may indicate that here to bypass the “Comments to the Author” section, enter your conflict of interest statement in the “Confidential to Editor” section, and submit your "Accept" recommendation.

Reviewer #2: All comments have been addressed

Reviewer #3: (No Response)

2. Is the manuscript technically sound, and do the data support the conclusions?

Reviewer #2: Yes

Reviewer #3: No

3. Has the statistical analysis been performed appropriately and rigorously? 

Reviewer #2: Yes

Reviewer #3: No

4. Have the authors made all data underlying the findings in their manuscript fully available?

Reviewer #2: No

Reviewer #3: Yes

5. Is the manuscript presented in an intelligible fashion and written in standard English?

Reviewer #2: Yes

Reviewer #3: Yes

6. Review Comments to the Author

Reviewer #2: Thank you to the authors for their thoughtful and thorough response to my comments. All of my comments have been addressed to my satisfaction. I have no further comments or concerns.

Reviewer #3: This paper reports on the feasibility of one part of a four part program designed to prevent depression in young people. While the focus of the Moving Stories intervention and the broader STORM program is admirable, I have several significant concerns regarding the manuscript. First, it is not clear to me why this is a stand alone paper - it seems to me that the authors may be salami slicing data from the larger RCT when it could have been presented as a single hybrid effectiveness-implementation trial. Further, the analyses used in this study are very simple and do not provide meaningful outputs e.g. simply reporting on costs does not form the basis of an economic analysis. It is difficult to judge if the intervention is cost effective or not as it is not compared to anything. Further, the description of the qualitative analysis is concerning - note taking rather than recording and transcribing verbatim limits the authenticity of the qualitative data, and there is no qualitative framework provided to the approach for analysis. I have some additional smaller concerns, but unfortunately, the ones outlined above are difficult to overcome. As such, I don't think this is suitable for publication in PLOS ONE.

7. PLOS authors have the option to publish the peer review history of their article (what does this mean?). If published, this will include your full peer review and any attached files.

Reviewer #2: No

Reviewer #3: No

---

## [Author Response · Author response to Decision Letter 1]

3 Nov 2021

Mandy Gijzen

Trimbos Institute

P.O. Box 725, 3500 AS, Utrecht, 

The Netherlands.

01-11-2021

Response to the Editor and the Reviewer

Ref: PONE-D-21-17344R1

Title: Feasibility of a serious game coupled with a contact-based session led by lived experience workers for depression prevention in high-school students

Journal: PLOS ONE

Dear Dr. Mordaunt,

We would like to thank you for the opportunity to respond to the comments of reviewer 3 to improve and further clarify our manuscript. The comments of the editor in italics; followed by our response, while changes made in the manuscript are between quotation marks. 

1. The study appears to present the results of original research. It appears to be an implementation/scaling of an approach developed in an earlier clinical trial.

2. Results do not appear to have been published elsewhere. The second reviewer has raised the question of "salami slicing" and it would be helpful for the authors to clarify what approach has been taken with regards to the publication of results from the underlying study. My interpretation is that this could be considered an "extension phase" of the original trial, or as the authors have called it, an implementation study focused on the scaling of the capability developed and studies in the original trial. I do think implementation science is a separate issue to clinical trials, but I think clearly distinguishing this is important for the record.

In regard to comments 1 and 2:

In short,

1. The preventive STORM program (consisting of several interventions) is currently being evaluated in an RCT

2. During the trial one preventive intervention -Moving Stories- proofed hard to execute as several obstacles were met during the trial.

3. This raised questions about the feasibility of rolling out Moving Stories in the future, once the STORM program effectiveness has been ascertained. 

4. Hence the need for a feasibility study to look into the barriers and facilitating factors for implementing Moving Stories 

Therefore, our study is

• Not an implementation study in the sense of STaRI (which has to be conducted once the interventions has been proven to be effective)

• Neither a hybrid implementation-evaluation study (with some amount of salami slicing on our part)

• But a feasibility study looking into the barriers and facilitating factors for Moving Stories’ future role out should STORM has been proven to be effective.

In our Abstract and Introduction we have added a clarification of the specific aim and intent of the study. 

In the title of our MS we used terms to describe to study’s methodology: originally Process Evaluation and after revision Feasibility of Scaled-up Implementation. Now we begin to see that these qualifiers were unhelpful and may have caused confusion about the purpose of our study. Hence, we have changed the title of the MS once again (please, see below) and tried to do a better job in clarifying the purpose of the MS in both the Abstract and Introduction (please, see below).

Changes made in manuscript:

Title: 

“Feasibility of a serious game coupled with a contact-based session led by lived experience workers for depression prevention in high-school students”

Abstract, page 2:

Lines 30-32:

“The current study was carried out to assess inhibiting and promoting factors for scaling up Moving Stories once its effectiveness has been ascertained.”

Lines 43-44:

“Bringing Moving Stories to scale in the high-school setting appears feasible, but will remain logistically somewhat challenging.”

Introduction, page 6, lines 118-127:

“Therefore, the primary aim of the current study was to assess the feasibility of implementing Moving Stories as a universal preventive intervention within the STORM program as this intervention was newly added within the STORM approach. More specifically, during the STORM trial we aimed to record (1) the number and percentage of high-school students that actively played Moving Stories (coverage rate), (2) the average number of playing days out of the five (participation rate), (3) how successful LEWs were in reviewing all five mental health first aid (MHFA) strategies during the debriefing sessions and to what extend students could identify depressive symptoms (fidelity), (4) what other themes were addressed and resonated with the students during the debriefing sessions, and how the students responded to the personal experiences of the LEWs.”

Methods, page 12, line 255:

“Data was collected during the STORM trial.”

3. Experiments, statistics, and other analyses are largely performed to a high technical standard and are described in sufficient detail. The second reviewer has raised the possibility of whether aspects such as the interviews being recorded with notes rather than with recordings/coding, constitute a critical flaw. In the context of this being an implementation study, I would put it to the authors to respond to this with regards to feasibility in particular. It has been commented on but given that we could anticipate that this might be a criticism post-publication, this may warrant expanding on.

We have added a clarification on the chosen method of recording in our Methods. 

Changes made in manuscript:

Methods, Data collection, page 12-13, lines 275-279:

“In total, 49 debriefing sessions were provided by 11 LEWs. Notes were taken of 18 debriefing sessions to record what the LEW and students discussed during the debriefing sessions: 8 during study year of 2017-2018 and 10 during 2018-2019 with at least one debriefing session per school (see Table 1). These notes were kept by four researchers to ensure that topics noted were not the results of reporter bias.”

Discussion, Strengths and Limitations, pages 29, lines 672-675: 

“Tape-recording would have legal implications as participants would be identifiable. Moreover, tape-recording has the disadvantage participants are less likely to discuss personal issues freely [52, 53].”

4. Conclusions are presented in an appropriate fashion and are supported by the data.

5. The article is presented in an intelligible fashion and is written in standard English.

6. The research meets all applicable standards for the ethics of experimentation and research integrity.

7. The article adheres to appropriate reporting guidelines and community standards for data availability, however I would suggest that upon clarifying that this is an implementation study rather than an extension phase of a clinical trial, that the authors consider reviewing the manuscript for adherence to the Standards for Reporting Implementation Studies (StaRI) Statement (https://www.equator-network.org/reporting-guidelines/stari-statement/). This is particularly helpful if the manuscript is later included in a systematic review or other form of meta-research, in to maximise the assessed quality of the study (not withstanding any critical problems).

Please find the STaRI checklist, as requested 

Once again, many thanks for allowing us to make these revisions.

On behalf of all authors,

Mandy Gijzen

---

## [Editor Report · Decision Letter 2]

5 Nov 2021

Feasibility of a serious game coupled with a contact-based session led by lived experience workers for depression prevention in high-school students

PONE-D-21-17344R2

Dear Dr. Gijzen,

We’re pleased to inform you that your manuscript has been judged scientifically suitable for publication and will be formally accepted for publication once it meets all outstanding technical requirements.

Kind regards,

Dylan A Mordaunt, MB ChB, FRACP, FAIDH

Academic Editor

PLOS ONE

Additional Editor Comments (optional):

Thank you for your submission. I am satisfied that the previous comments have been addressed adequately and that the manuscrupt now meets the criteria for publication.
---

## [Editor Report · Acceptance letter]

17 Nov 2021

PONE-D-21-17344R2 

Feasibility of a serious game coupled with a contact-based session led by lived experience workers for depression prevention in high-school students 

Dear Dr. Gijzen:

I'm pleased to inform you that your manuscript has been deemed suitable for publication in PLOS ONE. Congratulations! Your manuscript is now with our production department. 

Kind regards, 

on behalf of

Dr. Dylan A Mordaunt 

Academic Editor

PLOS ONE